# Determination of ubiquitin fitness landscapes under different chemical stresses in a classroom setting

David Mavor[1], Kyle Barlow[2], Samuel Thompson[1], Benjamin A Barad[1], Alain R Bonny[1], Clinton L Cario[2], Garrett Gaskins[2], Zairan Liu[1], Laura Deming[3], Seth D Axen[2], Elena Caceres[2], Weilin Chen[2], Adolfo Cuesta[4], Rachel E Gate[2], Evan M Green[1], Kaitlin R Hulce[4], Weiyue Ji[1], Lillian R Kenner[1], Bruk Mensa[4], Leanna S Morinishi[2], Steven M Moss[4], Marco Mravic[1], Ryan K Muir[4], Stefan Niekamp[1], Chimno I Nnadi[4], Eugene Palovcak[1], Erin M Poss[4], Tyler D Ross[1], Eugenia C Salcedo[4], Stephanie K See[4], Meena Subramaniam[2], Allison W Wong[4], Jennifer Li[5], Kurt S Thorn[6], Shane Ó Conchúir[7], Benjamin P Roscoe[8], Eric D Chow[6,9], Joseph L DeRisi[3,6], Tanja Kortemme[7], Daniel N Bolon[8], James S Fraser[7]*

[1]Biophysics Graduate Group, University of California, San Francisco, San Francisco, United States; [2]Bioinformatics Graduate Group, University of California, San Francisco, San Francisco, United States; [3]Howard Hughes Medical Institute, University of California, San Francisco, San Francisco, United States; [4]Chemistry and Chemical Biology Graduate Program, University of California, San Francisco, San Francisco, United States; [5]UCSF Science and Health Education Partnership, University of California, San Francisco, San Francisco, United States; [6]Department of Biochemistry and Biophysics, University of California, San Francisco, San Francisco, United States; [7]Department of Bioengineering and Therapeutic Sciences, California Institute for Quantitative Biology, University of California, San Francisco, San Francisco, United States; [8]Department of Biochemistry and Molecular Pharmacology, University of Massachusetts Medical School, Worcester, United States; [9]Center for Advanced Technology, University of California, San Francisco, San Francisco, United States

*For correspondence: jfraser@ fraserlab.com

**Competing interests:** The authors declare that no competing interests exist.

**Abstract** Ubiquitin is essential for eukaryotic life and varies in only 3 amino acid positions between yeast and humans. However, recent deep sequencing studies indicate that ubiquitin is highly tolerant to single mutations. We hypothesized that this tolerance would be reduced by chemically induced physiologic perturbations. To test this hypothesis, a class of first year UCSF graduate students employed deep mutational scanning to determine the fitness landscape of all possible single residue mutations in the presence of five different small molecule perturbations. These perturbations uncover 'shared sensitized positions' localized to areas around the hydrophobic patch and the C-terminus. In addition, we identified perturbation specific effects such as a sensitization of His68 in HU and a tolerance to mutation at Lys63 in DTT. Our data show how chemical stresses can reduce buffering effects in the ubiquitin proteasome system. Finally, this study demonstrates the potential of lab-based interdisciplinary graduate curriculum.

**eLife digest** The ability of an organism to grow and reproduce, that is, it's "fitness", is determined by how its genes interact with the environment. Yeast is a model organism in which researchers can control the exact mutations present in the yeast's genes (its genotype) and the conditions in which the yeast cells live (their environment). This allows researchers to measure how a yeast cell's genotype and environment affect its fitness.

Ubiquitin is a protein that many organisms depend on to manage cell stress by acting as a tag that targets other proteins for degradation. Essential proteins such as ubiquitin often remain unchanged by mutation over long periods of time. As a result, these proteins evolve very slowly. Like all proteins, ubiquitin is built from a chain of amino acid molecules linked together, and the ubiquitin proteins of yeast and humans are made of almost identical sequences of amino acids.

Although ubiquitin has barely changed its sequence over evolution, previous studies have shown that – under normal growth conditions in the laboratory – most amino acids in ubiquitin can be mutated without any loss of cell fitness. This led Mavor et al. to hypothesize that treating the yeast cells with chemicals that cause cell stress might lead to amino acids in ubiquitin becoming more sensitive to mutation.

To test this idea, a class of graduate students at the University of California, San Francisco grew yeast cells with different ubiquitin mutations together, and with different chemicals that induce cell stress, and measured their growth rates. Sequencing the ubiquitin gene in the thousands of tested yeast cells revealed that three of the chemicals cause a shared set of amino acids in ubiquitin to become more sensitive to mutation.

This result suggests that these amino acids are important for the stress response, possibly by altering the ability of yeast cells to target certain proteins for degradation. Conversely, another chemical causes yeast to become more tolerant to changes in the ubiquitin sequence. The experiments also link changes in particular amino acids in ubiquitin to specific stress responses.

Mavor et al. show that many of ubiquitin's amino acids are sensitive to mutation under different stress conditions, while others can be mutated to form different amino acids without effecting fitness. By testing the effects of other chemicals, future experiments could further characterize how the yeast's genotype and environment interact.

## Introduction

Protein homeostasis enables cells to engage in dynamic processes and respond to fluctuating environmental conditions (*Powers et al., 2009*). Misregulation of proteostasis leads to disease, including many cancers and neurodegenerative diseases (*Balch et al., 2008*; *Lindquist and Kelly, 2011*). Protein degradation is an important aspect of this regulation. In eukaryotes ~80% of the proteome is degraded by the highly conserved ubiquitin proteasome system (UPS) (*Zolk et al., 2006*). The high conservation of the UPS is epitomized by ubiquitin (Ub), a 76 amino acid protein post-translational modification that is ligated to substrate amine groups, including on Ub itself in poly-Ub linkages, via a three enzyme cascade (*Finley et al., 2012*).

Perhaps due to its central role in regulation, the sequence of ubiquitin has been extremely stable throughout evolution. Only three residues vary between yeast and human (96% sequence identity). This remarkable conservation implies that the UPS does not acquire new functions through mutations in the central player, Ub. Instead the evolution of proteins that add Ub to substrate proteins (E2/E3 enzymes), remove Ub (deubiquitinating enzymes, DUBs), or recognize Ub (adaptor proteins) combine to create new functions, many of which rely on various poly-Ub topologies (*Sharp and Li, 1987*; *Zuin et al., 2014*). The role of Lys48 linked poly-Ub in protein degradation (*Thrower et al., 2000*) appears to be universally conserved, but the functions of other linkages are more plastic. Although mass spectroscopy of cell lysates has shown that every possible poly-Ub lysine linkage exists within yeast cells (*Peng et al., 2003*), only the roles of Lys11 linked poly-Ub in ERAD (*Xu et al., 2009*) and Lys63 linked poly-Ub in DNA damage (*Zhang et al., 2011*) and endocytosis (*Erpapazoglou et al., 2014*) are well characterized in yeast. Both of these linkages are central to

stress responses, mirroring some of the established roles for non-Lys48 linkages in other organisms (*Komander and Rape, 2012*).

Given this central role in coordinating a diverse set of stress responses, perhaps the high sequence conservation of ubiquitin is not surprising. However, classic Alanine-scanning studies showed that ubiquitin is quite tolerant of mutation under normal growth conditions (*Sloper-Mould et al., 2001*). The high mutational tolerance of Ub was further confirmed using EMPIRIC ('extremely methodical and parallel investigation of randomized individual codons'), where growth rates of yeast strains harboring a nearly comprehensive library of all ubiquitin point mutations were assessed in bulk by deep sequencing (*Roscoe et al., 2013*). Subsequent studies revealed that many of the constraints on the Ub sequence are enforced directly by the E1-Ub interaction (*Roscoe and Bolon, 2014*); however, the surprisingly high number of tolerant positions remained unexplained.

To address the paradox of the high sequence conservation and mutational tolerance of ubiquitin, we posed the problem to the first year students in UCSF's iPQB (Integrative Program in Quantitative Biology) and CCB (Chemistry & Chemical Biology) graduate programs. Previous EMPIRIC experiments on HSP90 suggested that reducing protein expression could reveal fitness defects that are otherwise buffered (*Jiang et al., 2013*). Similarly, the high expression level of Ub might buffer some fitness defects. As an alternative approach to reducing expression, the buffering might also be exposed by chemical perturbations. Chemical-genetic approaches have been successful in elucidating protein function by assessing the context dependence of mutations on organismal fitness (*Hietpas et al., 2013*; *Stockwell, 2000*). There is substantial evidence to link the pool of free Ub to the eukaryotic stress responses to many chemicals. For example, Ub overexpression is protective against the general translational inhibitor cycloheximide (*Hanna et al., 2003*) and the deubiquitinating enzyme Ubp6 is upregulated upon stress to increase the pool of free Ub in the cell (*Hanna et al., 2007*). Collectively, these results suggest that the sequence of Ub is subject to many constraints arising from interacting with diverse proteins while mediating the stress responses to distinct chemical perturbations.

The students performed the bulk competition experiments, deep sequencing and data analysis as part of an 8-week long research class held in a purpose-built Teaching Lab. In small teams of 4–5 students working together for 3 afternoons each week, they each examined a chemical stressor: Caffeine, which inhibits TOR and consequently the cell cycle (*Reinke et al., 2006*; *Wanke et al., 2008*); Dithiothreitol (DTT), which reduces disulfides and induces the unfolded protein response (*Frand and Kaiser, 1998*) and the ER associated decay (ERAD) pathway (*Friedlander et al., 2000*); Hydroxyurea (HU), which causes pausing during DNA replication and induces DNA damage (*Koç et al., 2004*; *Petermann et al., 2010*); or MG132, which inhibits the protease activity of the proteasome (*Jensen et al., 1995*; *Rock et al., 1994*). We expected MG132 to desensitize the yeast to deleterious mutations in Ub, as the inhibition acts on the final degradation of UPS substrates. For the other three chemicals we expected that specific sites on Ub would become sensitized to mutation. These sites could represent important Ub/protein binding interfaces that are required for Ub to bind to adaptor proteins and ligation machinery required to respond to a specific stress. Furthermore, we expected that Caffeine induced stress would be mediated through Lys48 linked poly-Ub (cell cycle), DTT induced stress would be mediated through Lys11 linked poly-Ub (ERAD), and HU induced stress would be mediated through Lys63 linked poly-Ub (DNA damage response).

Our data collectively show that stress reduces a general buffering effect and unmasks a shared set of residues that become less tolerant to mutation. Additionally, we have identified a small set of mutations that are specifically aggravated or alleviated by each chemical. Small fitness defects that are undetectable by EMPIRIC may still be significant over evolutionary timescales and are likely to similarly be enriched for certain chemicals (*Boucher et al., 2014*). We suggest that expanding the set of environmental stresses, and improved measurement of smaller fitness defects, might be able to explain the high sequence conservation of ubiquitin, as different positions in the protein are important for interactions mediating the specific responses to a wide variety of perturbations.

## Results

### Library construction

Previously, the fitness landscape of Ub in yeast was determined using eight competition experiments using the EMPIRIC strategy of deep sequencing short regions of all possible single amino acid substitutions during a growth competition experiment in rich media (*Roscoe et al., 2013*). These experiments measured all point mutants contained in short 30 base pair (bp)/10 amino acid residue stretches of the Ub open reading frame (ORF), which necessitated 8 separate competition experiments. To increase the throughput and reduce the cost of the experiment, we designed a barcoding strategy (*Fowler et al., 2014*), that allowed us to determine allele fitness in a single experiment using EMPIRIC with barcodes (EMPRIC-BC). We synthesized eighteen bp random barcodes (N18 BCs), which were ligated upstream of the Illumina sequencing primer binding site. The specific association of each unique N18 BC with a given mutant Ub allele was then established through paired end sequencing of the Ub ORF and the N18 BC (*Figure 1A*) The resulting lookup table of BCs and alleles was then employed in our competition experiments to count alleles by directly sequencing the N18 BCs. In addition to simplifying the experiment, this strategy enabled us to count the alleles with a short, single end sequencing run, substantially reducing cost. The library is ~97.5% complete at the amino acid level. We observed a slight GC bias in the codon coverage (*Figure 1B–C*), which is likely due to the cloning method that initially generated the Ub mutants (*Hietpas et al., 2012*). Most substitutions are associated with many N18 BCs, with a median of fifteen unique barcodes representing a specific amino acid substitution (*Figure 1D*).

### Determining the Ub fitness landscape in DMSO

To determine the differential fitness landscape of Ub under different chemical stresses, we first conducted an EMPIRIC-BC experiment under 0.5% DMSO to serve as a control (*Figure 2*). The resulting fitness landscape is quite similar to the previously published dataset, which was collected under no chemical stress (*Roscoe et al., 2013*) (*Figure 3A*). The lowest fitness scores occurred at premature stop codons and residues that are critical to build Lys48 poly-Ub linkages (Lys48, Ile44, Gly75, Gly76). As previously observed, much of the protein surface is tolerant to mutation. Based on the average value of the stop codon substitutions, we set a minimum fitness score of -0.5 (*Figure 3B*). Comparisons of biological replicates indicated that the data were reproducible and well fit by a Lorentzian function centered at 0, as expected for the ratio of two Gaussian distributions (*Figure 3C,D*).

### Chemical perturbations sensitize many positions in Ub

Because our competition experiments require cells growing for multiple generations in log phase, we conducted our experiments at chemical concentrations that inhibit yeast growth by 25%. These chemical concentrations are not as high as used in previous transcriptional studies of yeast chemical stress responses (*Gasch et al., 2000*). For Caffeine (7.5 mM), and DTT (1 mM) we determined the IC25 for each drug by following growth via optical density (Figure 4—figure supplement 1). Since HU (25 mM) induced a lag phase followed by near wild type like growth, we determined the IC25 concentration by monitoring yeast growth from two to five hr post treatment. DMSO (0.5%) and MG132 (50 uM) did not inhibit growth.

Next, we performed the EMPIRIC-BC experiment with each chemical perturbation (*Figure 4B*). In Caffeine, DTT, and HU (*Figure 4A–D*, *Figure 4—figure supplements 1–3*) many mutations are sensitized, and become less fit than in DMSO. Generally this increased sensitivity is localized around the C-terminus, which is essential for E1 activation, and the hydrophobic patch, which is the dominant interface for protein-protein interactions.

To compare the responses to each perturbation, for each pairwise comparison we plotted the fitness scores for each mutant as a scatter plot and calculated the residual to the identity line. We compared the distribution of these residuals to the distribution of residuals calculated by the DMSO self comparison (*Figure 5*). Caffeine, DTT and HU generally sensitize the protein to mutation, which is evident in the enrichment of mutations with reduced fitness compared to the DMSO self distribution. These newly sensitized mutations are largely shared between these different chemical perturbations.

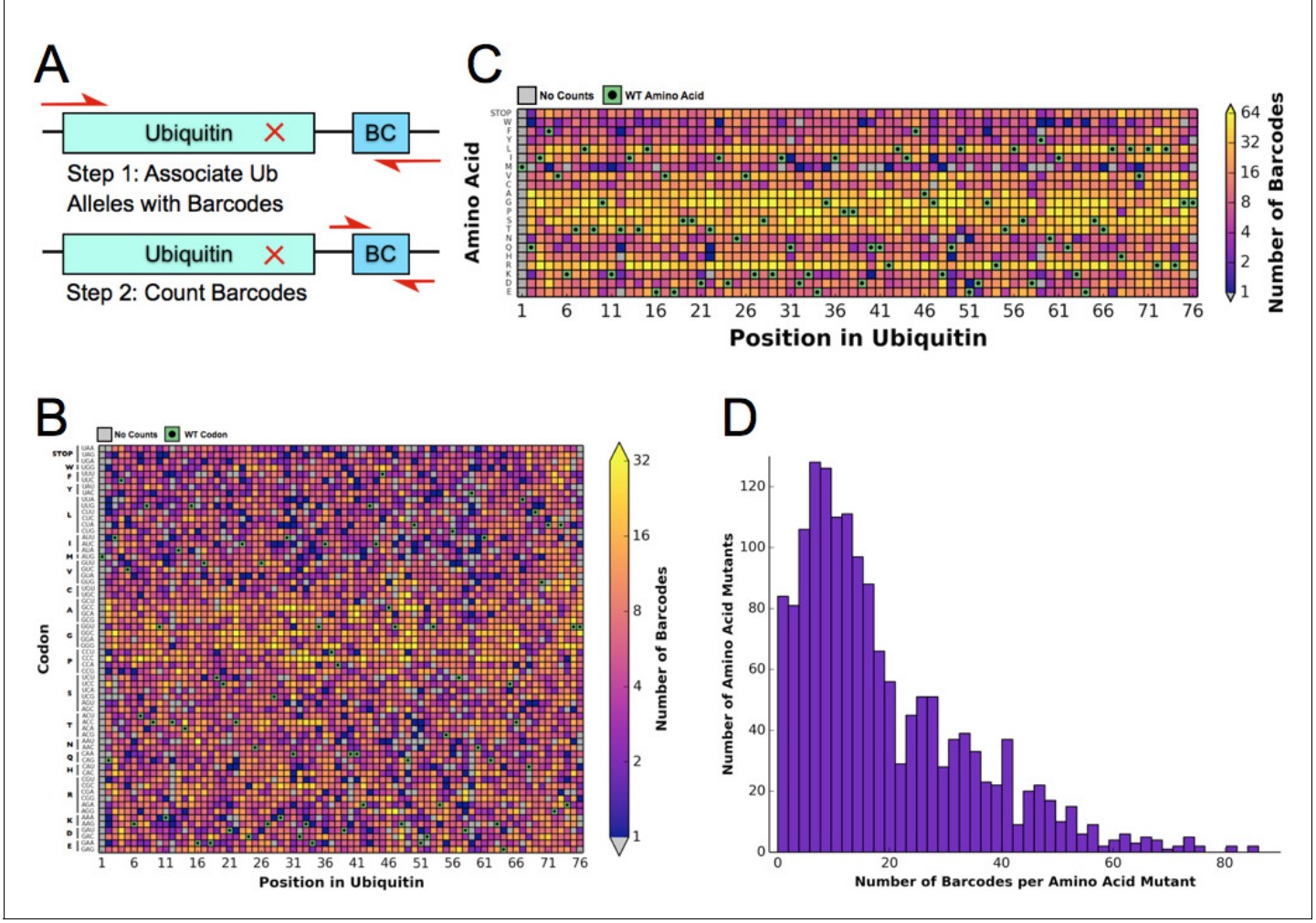

**Figure 1.** Barcoding enables a bulk competition experiment of ~1500 Ubiquitin variants. (**A**) Prior to the competition experiment, ubiquitin alleles were specifically associated with unique barcodes through a paired end sequencing. To monitor the frequency of different alleles during the competition experiments, we directly sequenced the barcodes in a short single end read. (**B**) The library contains most codon substitutions and almost all are associated with multiple barcodes. A slight GC bias is seen in the cloning. WT codons are shown in green and missing alleles are shown in grey. (**C**) The amino acid coverage of the library is almost complete. WT residues are shown in green and missing alleles are shown in grey. (**D**) Examining the number of barcodes per amino acid substitution shows that 2.5% of the library is missing and the median number of barcodes per substitution is 15.

In contrast to the sensitizing effects of DTT, Caffeine, and HU, the proteasome inhibitor MG132 increases mutational tolerance throughout the protein. This effect can be seen in the slight shift of the residuals distribution to the right when compared to the DMSO self distribution (*Figure 5D*). The effect is small at the MG132 concentration we assayed, which is likely due to the poor penetrance of MG132 in yeast cells containing a wild type allele of *ERG6* (*Lee and Goldberg, 1996*). This alleviating interaction is likely because MG132 directly perturbs proteasome, reducing the impact of defects related to Lys48 linked poly-Ub chains and leaving functions related to other, non-degradative poly-Ub topologies unperturbed. Also, reducing proteasome activity may increase the free pool of Ub in the cell by reducing the number of Ub proteins degraded by the proteasome. This increased pool of Ub could buffer the effects of deleterious Ub mutants participating in non-proteasomal functions. The consequence of an increased pool of free Ub might therefore lead to the general alleviating interactions observed between Ub mutants and MG132 treatment.

Additionally, the difference distributions are wider than the distribution between DMSO replicates. This result shows that the perturbations unmask previously buffered fitness defects that are phenotypically important for each perturbation. If the biological role of each residue were

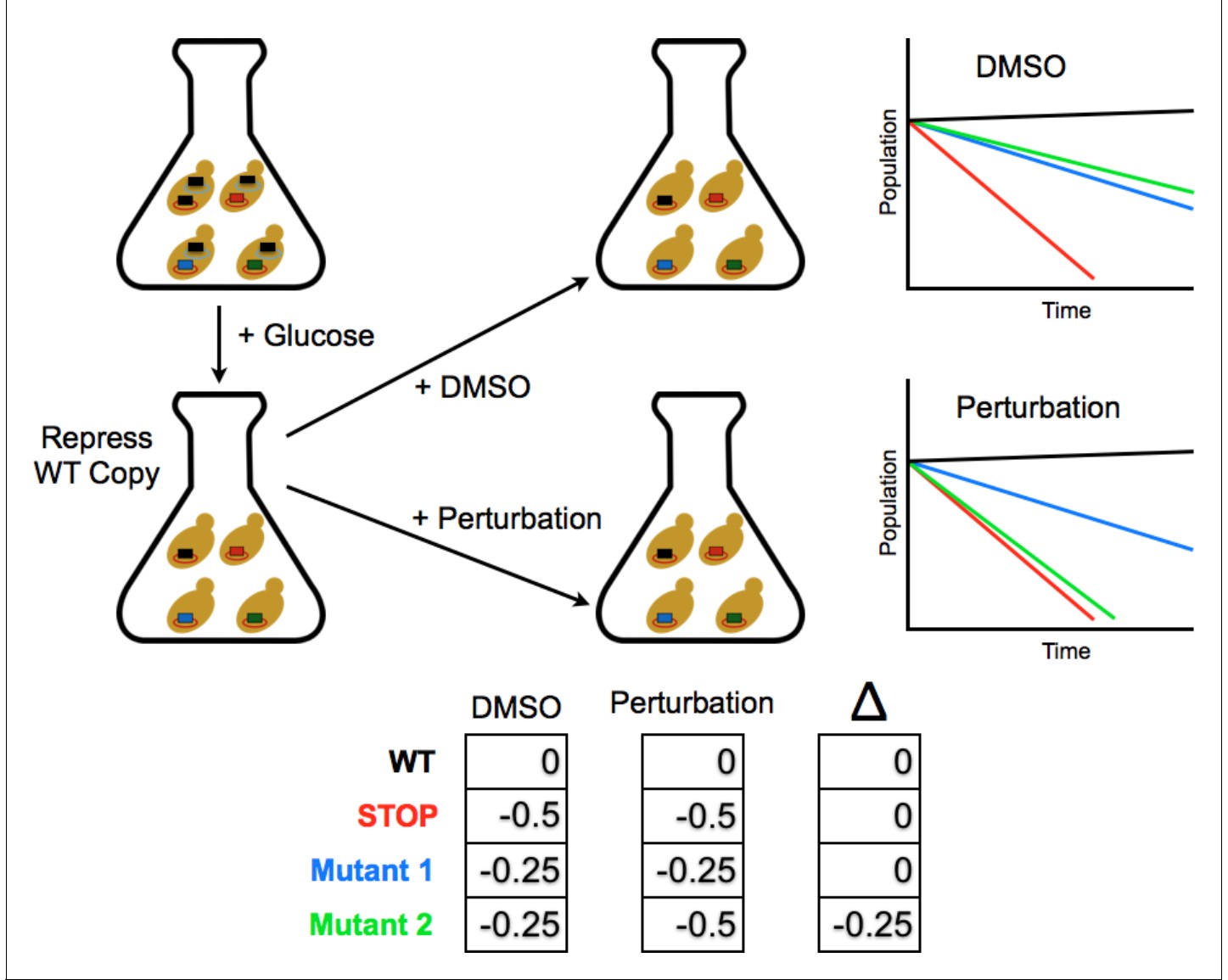

**Figure 2.** Competition experiment based on a galactose inducible Ub. The fitness of all ubiquitin mutants was measured in a single culture by shutting off the galactose-driven wild type copy. This allows a constitutively expressed mutant to be the sole source of ubiquitin for the cell. The library was grown for 48 hr in galactose to remove dominant negative alleles and then expression of the wild type copy repressed by the addition of glucose. Upon repression of the wild type copy, chemical perturbations were added and the yeast were grown for multiple generations. Fitness scores were calculated for each mutant based on the relative frequencies of mutant and wild type alleles over multiple generations. The ratio of (mutant counts): (wild type counts) was computed for each time point and a line fit to these ratios vs. generation time. The fitness score is the slope of the linear fit.

independent of perturbation then the distributions would be shifted without affecting the shape. Instead these data show that each perturbation uncovers unique roles of Ub residues in responding to a specific perturbation.

## Rosetta ΔΔG modeling indicate that sensitive mutants mildly perturb stability

One potential explanation of the buffer unmasked by the chemical perturbations is the stability of the Ub protein itself. Although Ub is highly stable (*Ibarra-Molero et al., 1999*; *Wintrode et al., 1994*), mutations that destabilize it may lead to misfolding or perturb Ub/protein interactions important for UPS function. To assess the degree to which mutational destabilization of ubiquitin itself is

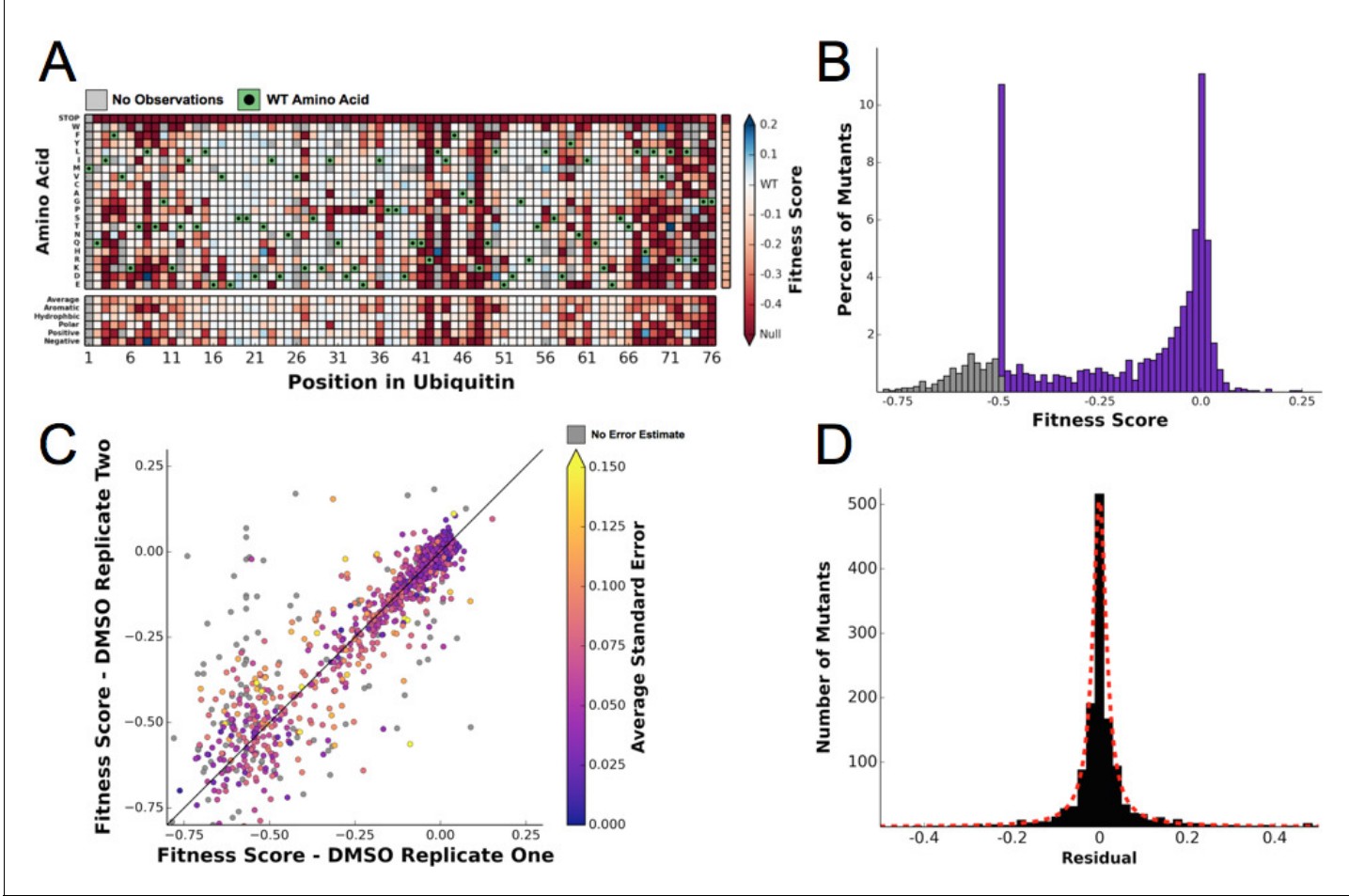

**Figure 3.** Ubiquitin fitness scores determined in DMSO are replicable and define the 'unperturbed' Ub fitness landscape. (**A**) Heatmap showing the fitness of observed ubiquitin alleles. Scores presented are the average of three biological replicates. Wild type amino acids are shown in green and mutations without fitness values (due to lack of barcode or competition sequencing reads) are shown in grey. The average fitness score of each position and the averages of substitutions binned by amino acid characteristics are shown below. The single column on the far right shows the average of each amino acid substitution across all positions. (**B**) The distribution of fitness values is shown and colored based on fitness score. Grey bins reflect fitness scores that were reset to -0.5. (**C**) Biological replicates of the competition experiment in DMSO are well correlated ($R^2$ = 0.79). Each point represents the fitness score of a mutant in two biological replicates. Points are colored based on the average standard deviation of the barcodes contributing to each fitness score. (**D**) The distribution of the residuals to the identity line between two DMSO replicates is symmetric and well modeled by a Lorentzian ($X_0$ = 0, $\Gamma$ = 0.0035, scaled by 1600).

The following figure supplements are available for figure 3:

**Figure supplement 1.** Error estimates for fitness scores determined in DMSO.

**Figure supplement 2.** Fitnesses determined in DMSO are well correlated to the previously determined unperturbed fitnesses.

predictive of a decrease in mean fitness for each perturbation, we used the macromolecular modeling software Rosetta to estimate changes in protein stability (*Kellogg et al., 2011*; *Kortemme and Baker, 2002*) for every mutation in our library. With the resulting predictions, we classified each ubiquitin mutation as either destabilizing (change in Rosetta Energy Units (REU) > = 1.0) or neutral/stabilizing (change in REU < 1.0). We observed a significant difference in experimental fitness between the two predicted classes for all conditions (*Figure 6*). This result holds independently of the absolute mean experimental fitness score of each perturbation, meaning that the difference in mean experimental fitness between predicted destabilizing and neutral mutations is not simply the result of lower mean destabilizing fitness scores. These results suggest that ubiquitin stability is

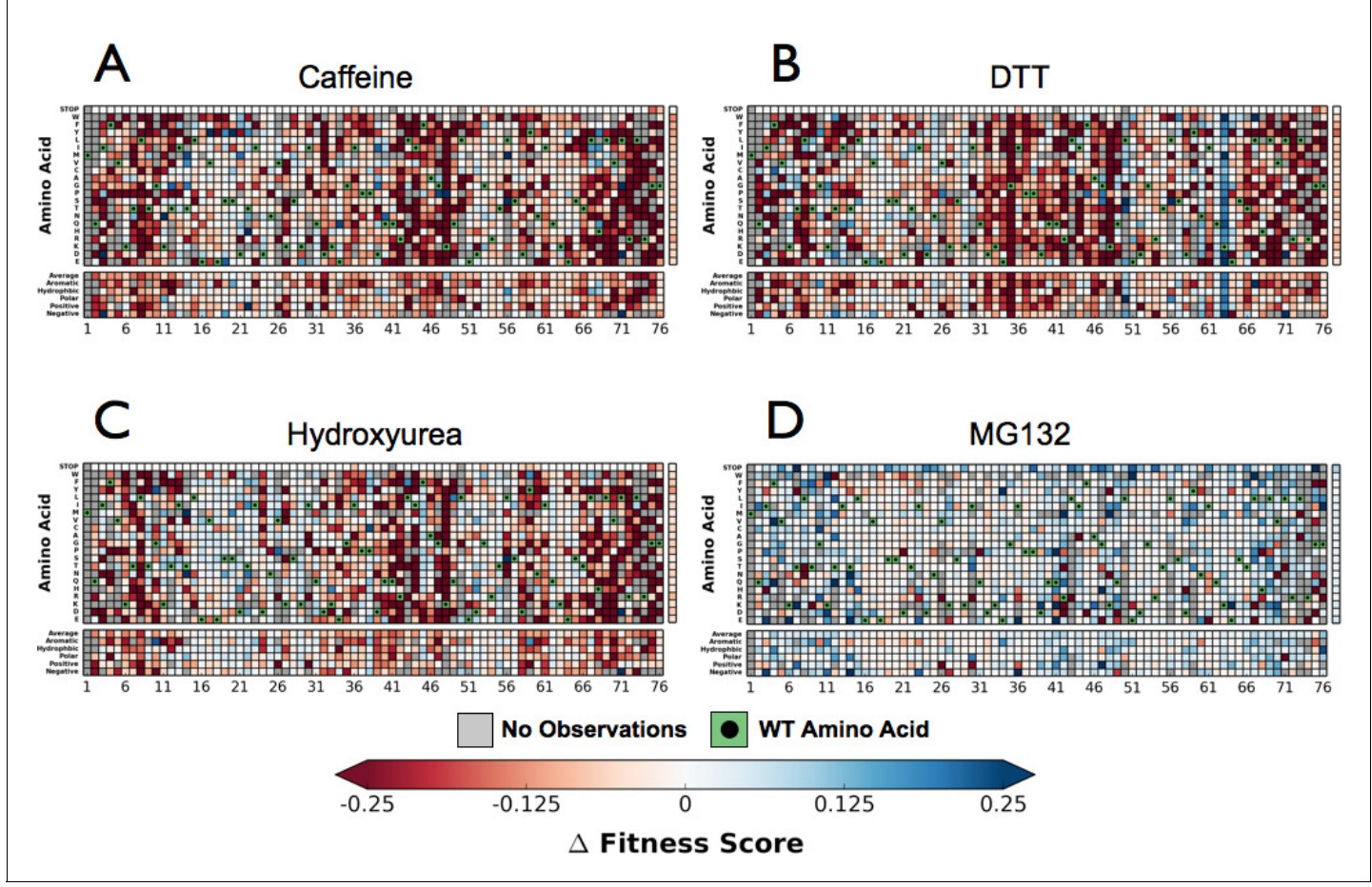

**Figure 4.** Perturbations sensitize ubiquitin to mutations. The difference in fitness between DMSO and a perturbation for each Ub allele: (A) Caffeine, (B) DTT (C) Hydroxyurea (D) MG132. Wild type amino acids are shown in red and mutations without fitness values (due to lack of barcode or competition sequencing reads) are shown in grey.

The following figure supplements are available for figure 4:

**Figure supplement 1.** Perturbations sensitize ubiquitin to mutations.

**Figure supplement 2.** Perturbations sensitize ubiquitin to mutations.

**Figure supplement 3.** Growth curves.

more important for fitness in each of the perturbed conditions than in unperturbed yeast. Under stress, subtle changes in Ub stability could induce fitness defects that are otherwise buffered under control (DMSO) conditions. Furthermore, even small changes to ubiquitin stability could induce considerable changes to the Ub conformational ensemble that could destabilize Ub/protein complexes (*Lange et al., 2008*; *Phillips et al., 2013*). Adaptability within the UPS could buffer these defects in DMSO, but they can be revealed upon chemical stress.

## Mutational sensitivity is primarily localized to three regions of Ub

To assess the role of specific positions in Ub we averaged the fitness score of each amino acid mutation at a given position. We then binned each position into sensitive ($\leq$-0.35), intermediate (-0.35 to -0.075) and tolerant ($\geq$-0.075) and examined the distribution of average fitness in each condition (*Figure 8A*). These distributions again show that most positions in Ub are tolerant to mutation in DMSO, but many positions are sensitized upon chemical perturbation.

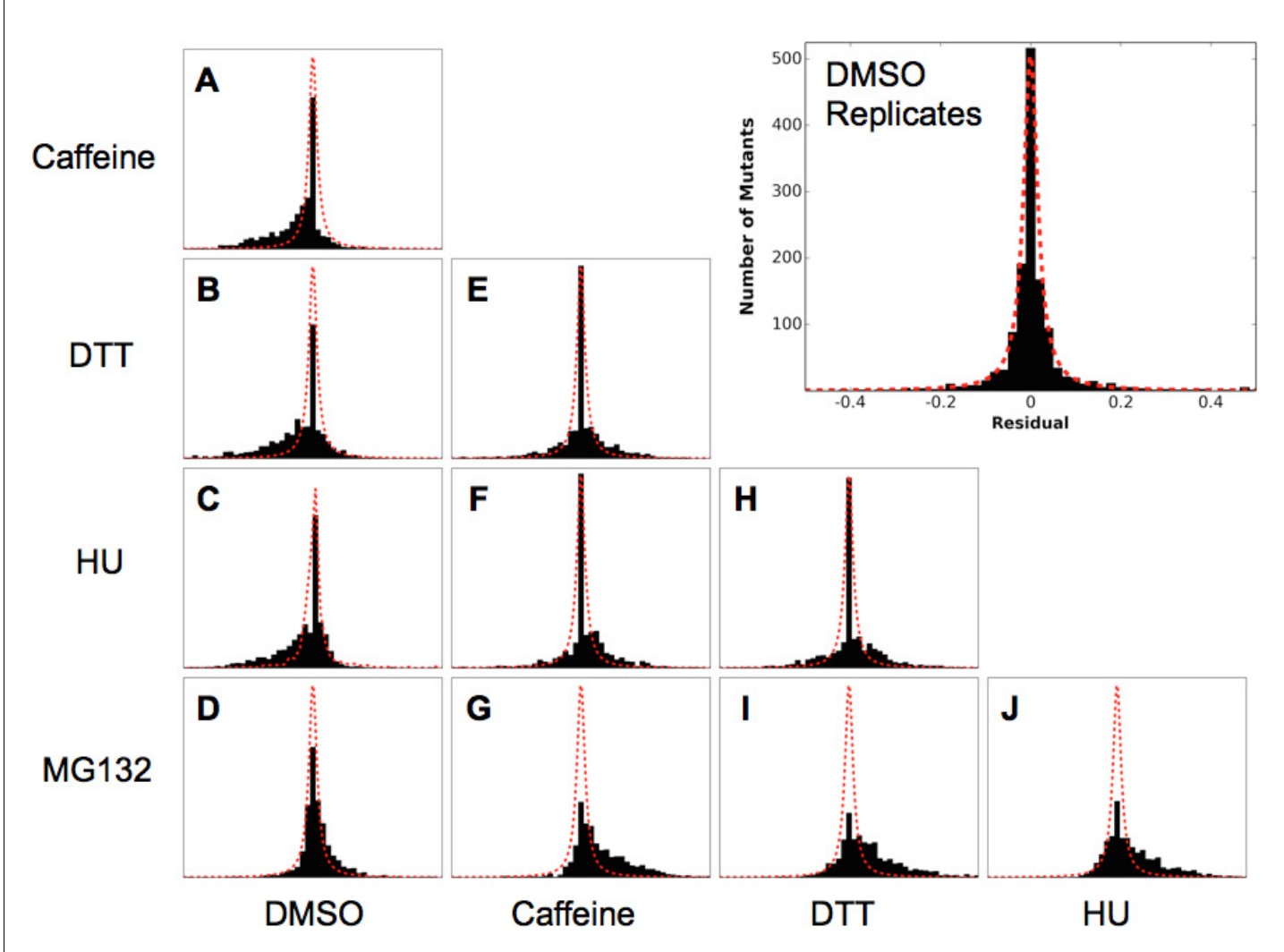

**Figure 5.** Residual distributions highlight a shared mutational response between Caffeine, DTT and HU. The residuals between datasets shows are shown with the Lorentzian representing the biological replicates of DMSO in red. When compared to DMSO, three perturbations (Caffeine, DTT and HU) shift the distributions to the left, which highlights the increased sensitivity to mutation. In contrast, MG132 slightly shifts the distribution to the right, which highlights the alleviating interaction between MG132 and deleterious ubiquitin alleles. Comparisons between Caffeine, Hydroxyurea and DTT are symmetric but with longer tails than the control experiments. This result suggests a shared response comprised of many sensitized residues and a smaller number of perturbation-specific signals.

In DMSO only residues with well-established roles are sensitive: Arg42 (E1 activation), Ile44 (hydrophobic patch hotspot), Lys48 (essential Lys48 linked poly-Ub) and Gly75-Gly76 of the C-terminus (E1 activation) (*Figure 7*). The face opposite the hydrophobic patch is mostly tolerant and the protein core and residues adjacent to the sensitive residues are mostly intermediate (*Figure 8B* - i). When treated with Caffeine, DTT or HU, a shared set of residues become sensitive (*Figure 8B* ii– iv, *Figure 8C*). These residues are either: located adjacent to DMSO sensitive residues (e.g. Leu73, which is in the C-terminal tail); residues with important biological functions that of intermediate sensitivity in DMSO (e.g. Leu8, Val70, which are important hydrophobic patch residues); or core residues (e.g. Ile36, Leu71). These positions tolerated a small set of substitutions in DMSO but upon perturbation became only tolerant of mutations that share physical chemistry with the wild type residue.

Examining the positions made intermediate by the perturbations highlights the similarities and differences between the DTT and Caffeine/HU datasets (*Figure 8D*). All three perturbations shift a shared set of residues to the intermediate bin. These residues are mostly surface residues on the

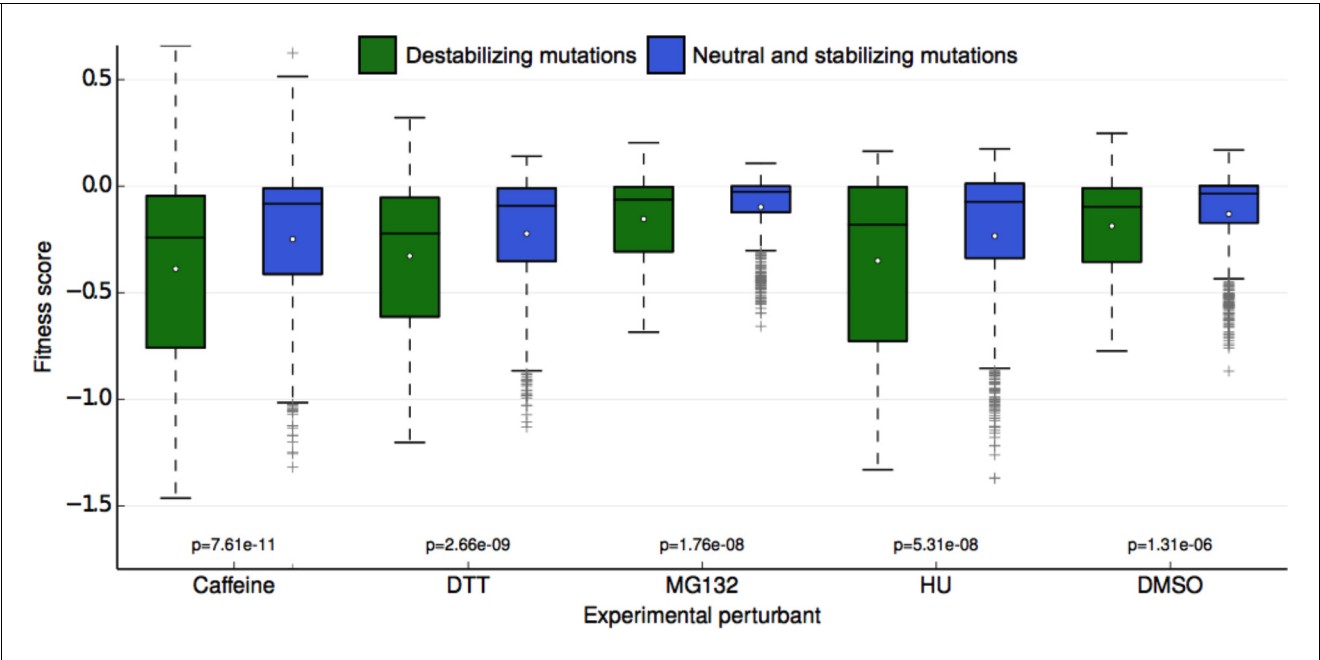

**Figure 6.** Fitness score data binned by Rosetta stability predictions. Fitness scores for each of the 5 sets of experimental conditions are shown along the y-axis as boxplots. Scores are grouped first by their respective experimental condition, and then by the change in stability of the ubiquitin monomer of the mutation estimated by Rosetta. Mutations that Rosetta predicts to be neutral or stabilizing (REU (Rosetta Energy Units) < 1.0) are shown in blue boxes; mutations predicted to be destabilizing (REU >= 1.0) are shown in green boxes. The mean of each fitness score distribution is shown as a white dot. The p-value of the two-sided T test between the fitness mean of mutations predicted to be stabilizing and those predicted to be neutral/stabilizing is shown at the bottom of the plot. Experimental conditions are arranged from left to right along the x-axis in order of decreasing p-value.

tolerant face of Ub. In DMSO they tolerate a wide range of amino acids. Upon perturbation the mutational tolerance is reduced to amino acids generally compatible with surface residues. For example in DMSO, Asp32 is tolerant to any substitution except Proline. Upon perturbation, this position is restricted to polar and negative substitutions.

Additionally, DTT uniquely shifts five positions into the intermediate bin. This is due to subtle changes in the tolerance of positions that are otherwise highly tolerant. For example, mutations at Arg54 are well tolerated in all other conditions. However, in DTT mutations to negative residues become deleterious while all other substitution remain tolerated. This suggests that Arg54 may participate in a salt bridge during a protein-protein interaction that is involved in mediating the cellular response to DTT.

We also uncovered newly tolerant positions, which are uniquely tolerant to each of the perturbations (*Figure 8E*). These positions tend to be mildly sensitive to most mutations in DMSO, suggesting that these residues are involved in biological pathways that are important for cellular function, but not essential. When perturbed, these positions are mildly desensitized to mutation, with little regard for mutant amino acid identity. The most striking example is at Lys63 in DTT. In all other conditions any mutation of this residue is mildly deleterious. Because Lys63 linked poly-Ub chains are important for efficient cargo sorting in the endosome, this sensitivity is likely due to an endocytic defect. However, upon DTT treatment most mutants of Lys63 show WT-like fitness. This suggests an epistatic interaction between DTT treatment and mutations at Lys63. The endocytic defect caused by mutation Lys63 is likely masked by a similar endocytic defect induced by DTT treatment. Average difference maps showing the (DMSO - perturbation) fitness score highlight the features that underlie the sensitized and desensitized positions (*Figure 4*).

In a final effort to resolve the dichotomy between the Ub fitness landscape and the evolutionary record, we visualized the average fitness of each position in DMSO and compared it to the minimum of the average fitness of each position for all perturbations (*Figure 8F,G*). The data in DMSO again shows that biologically relevant positions are sensitive, the face opposite the hydrophobic patch is

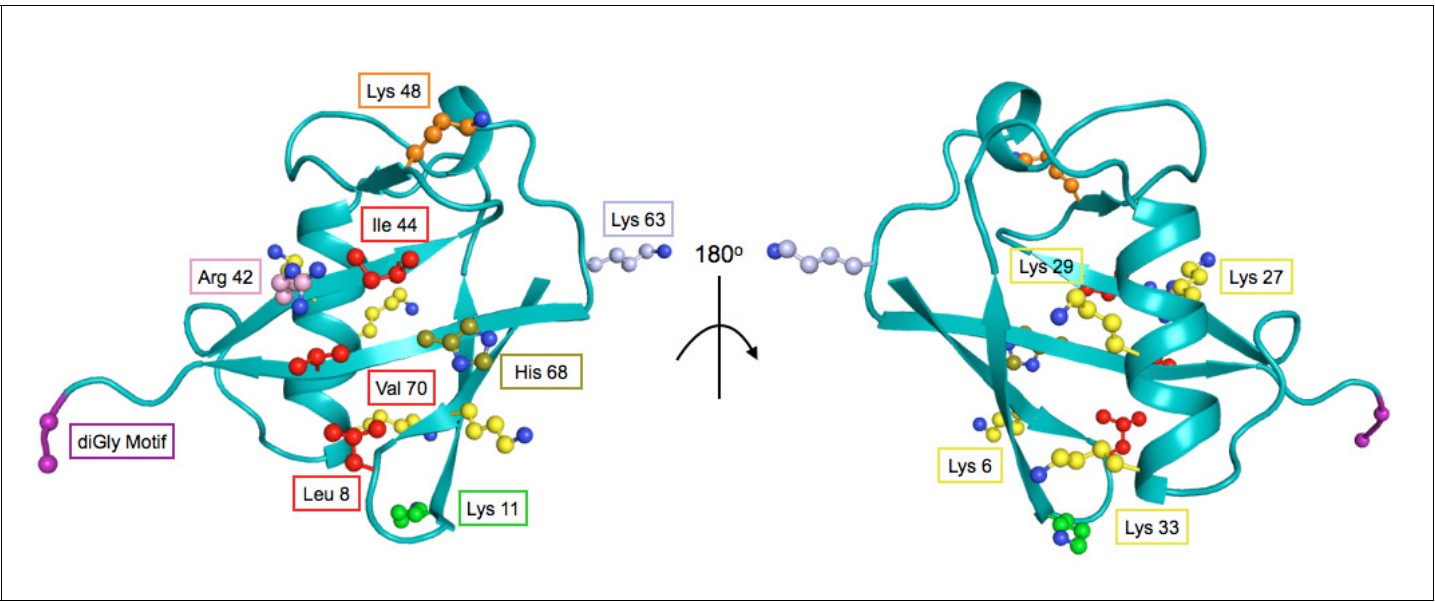

**Figure 7.** The structure of Ub highlighting important residues. Cartoon model of Ub (PDB 1UBQ) with important residues colored as follows: Lys48 - orange, Lys63 - light blue, Lys11 - green, other Lys residues - yellow, hydrophobic patch (Leu8, Val40, Ile 44) - red, C-terminal diGly motif (Gly75 and 76) - purple, Arg42 - pink, His68 - olive.

extremely tolerant to mutation, and that core residues are intermediately tolerant. Perturbations dramatically increase mutational sensitivity at the C-terminus, around the hydrophobic patch and at some core positions. However, much of the tolerant face of the protein remains tolerant to mutation in all of the perturbations. By exploring a wider array of perturbations we should be able determine the environmental pressures that constrain these tolerant positions and explain the extreme conservation of Ub.

## Specific elements of the shared response to perturbations

To determine the elements of the shared response to HU, Caffeine and DTT, we defined 'shared sensitizing mutations' as those that were both sensitizing (delta fitness ≤ -0.2 for all perturbations) and consistent between perturbations (within 0.1 of the regression line) (*Figure 9A* and *Figure 9— source data 1*). Most of these mutations change from being mildly deleterious to being nearly null upon chemical stress. For example, in DMSO Ub tolerates mutation to small hydrophobics and other polar residues at Thr7. However, chemical stresses causes mutations of small hydrophobic or charged residues at this position to be deleterious. As Thr7 is adjacent to the hydrophobic patch residue Leu8, this sensitization is likely due to non-polar substitutions disrupting Ub adaptor protein binding and poly-Ub packing (*Komander and Rape, 2012*). Additionally, typically destabilizing substitutions such as Proline or Tryptophan generally become more deleterious under perturbation.

## Specific residues connect different stresses to Ub protein-protein interactions

We also investigated specific signals outside of the shared sensitizing response (*Figure 9B*). We identified perturbation specific mutations by comparing the change in fitness scores of each of the sensitizing perturbations (*Figure 9—source data 2*, *3*). Because these mutants are not sensitized by all of the perturbations they likely alter binding to specific adaptors, conjugation machinery, or substrates. Most of these perturbation specific mutants are tolerated in Caffeine and HU, but sensitized by DTT treatment. However, the H68Y mutation differs as DTT and HU treatments sensitize this mutation whereas Caffeine treatment does not. His68 is an important position found at the interface between Ub and adaptor domains such as UIM and UBA domains. These domains are important for

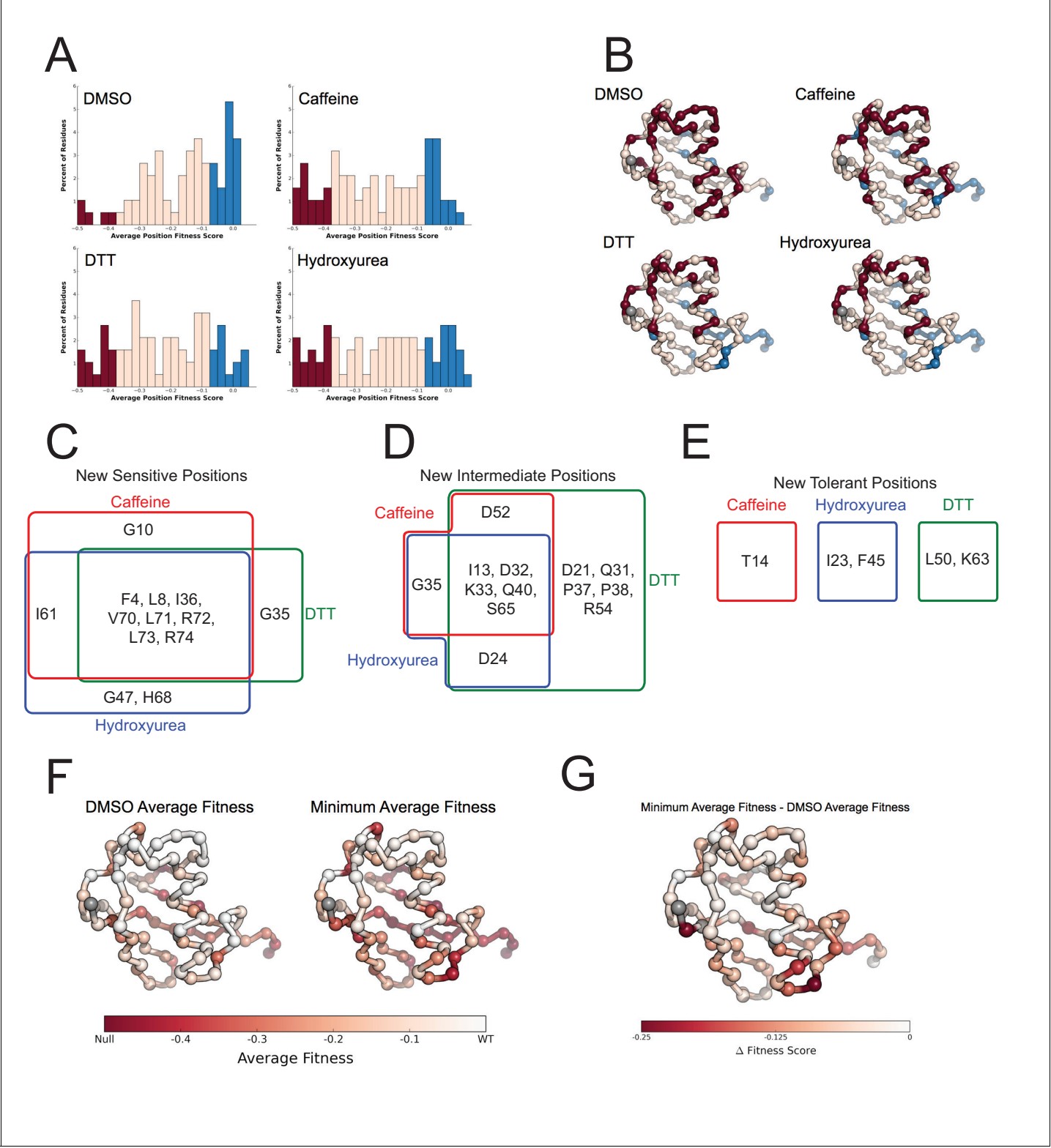

**Figure 8.** Average fitness values show sensitization by the perturbations at each position in ubiquitin. (**A**) Based on the average fitness score, positions were binned into tolerant (>=-0.075 - Blue), intermediate (<-0.075 to > -0.35 - Pink) and sensitive (<= -0.35 - Red). (i) DMSO (ii) Caffeine (iii) DTT (iv) Hydroxyurea show a shift from tolerant to intermediate and sensitive positions. (**B**) Positions binned by average fitness score mapped onto the ubiquitin structure. C-alpha atoms are shown in spheres and the residues are colored as in **A**. Met1 is colored grey. (**C**) New sensitive positions induced by the perturbation describe a shared response to perturbation with 8 of 13 positions shared between Caffeine, DTT and HU. (**D**) New intermediate positions

*Figure 8 continued on next page*

*Figure 8 continued*

highlight the similarity between HU and Caffeine, with DTT sensitizing a unique set of residues. (E) New tolerant positions are unique to each perturbation. (F) Average position fitness scores mapped onto ubiquitin. (i) DMSO (ii) Minimum average fitness score in all perturbations. C-alpha atoms are shown in spheres and the residues are colored according to average fitness. Met1 is colored grey. (G) Minimum average fitness scores – DMSO average fitness scores mapped onto ubiquitin. C-alpha atoms are shown in spheres and the residues are colored according to the difference in fitness. Met1 is colored grey. With this small set of perturbations most positions are sensitized.

the trafficking of ubiquitinated proteins. His68 lies adjacent to the hydrophobic patch and binding to UIM domains is reduced when it is protonated (*Fujiwara et al., 2004*). In contrast, when His68 is mutated to Val in Ub, the binding to UIM domains is increased, likely mimicking the deprotonated state that forms a hydrophobic surface (*Fujiwara et al., 2004*).

Lys11 is similarly important for Ub biology and shows a specific sensitization to DTT. Lys11 linked poly-Ub chains are the second most abundant linkage in yeast. These chains likely signal for degradation by the proteasome, like Lys48 linked chains, and have been implicated in the response to ER stress (*Xu et al., 2009*). In DMSO all substitutions, except to negative and aromatic residues, are tolerated. However, substitutions to Leu, Ile, His and Asn are sensitized uniquely in DTT. These data suggest that Lys11 is mediating an interaction to DTT induced stress. Although previous studies have indicated a synthetic lethal interaction between Lys11Arg and DTT (*Xu et al., 2009*), in our experiments, at lower DTT concentrations, the relatively high fitness of Lys11Arg suggests that the structural role of the positively charged residues and not poly-Lys11 Ub linkages may dominate the physiological response.

In addition to fitness defects that are likely due to perturbing Ub/protein interfaces, we also observed defects due to perturbing ploy-Ub chain structure and dynamics. Lys63 linked poly-Ub chains exist in three distinct conformations in solution (*Liu et al., 2015*). The populations of these conformational states help determine binding partner selection between Lys63 linked chains and adaptor proteins. Mutating Glu64 to Arg biases the chains towards the open conformation (*Liu et al., 2015*). In DMSO, the mutation of Glu64 to a positive residue caused a fitness defect. In Caffeine and HU these mutants are sensitized and the fitness defect is further increased. However, DTT treatment increased the tolerance to positive mutations at this position, again suggesting an interaction between Lys63 linked poly-Ub and DTT treatment.

## Discussion

We have determined the fitness landscape of Ub in yeast grown in the presence of five chemical perturbations. We identified newly sensitized positions in the protein, which supports the hypothesis that the Ub sequence is highly constrained by its role in a wide array of environmental stress responses. Although each perturbation had some unique features, we observed a general buffering effect that may have obscured mutational sensitivity in the previously determined Ub fitness landscape.

Perhaps the most surprising result in our study was the failure to recapitulate the synthetic lethal interaction between Lys11Arg and DTT (*Xu et al., 2009*). This interaction was observed using the same strain (SUB328), however fitness was determined through a dilution spot assay on an agar plate containing 30 mM DTT. Our experiments were conducted in liquid culture with 1 mM DTT refreshed every sampling period. It is likely that we did not achieve a stress regime where Lys11 poly-Ub is essential for DTT tolerance. The Lys11Arg mutation induces the upregulation of proteins involved in ERAD including Ubc6, the ERAD E2. Also, the turnover of known ERAD substrates is unaffected by the Lys11Arg mutation, suggesting that Lys48 linked chains can be substituted for Lys11 linked chains (*Xu et al., 2009*). These adaptations could be sufficient to counteract the loss of Lys11 poly-Ub in our experiments, but are insufficient at higher concentrations of DTT. It would be interesting to explore these two regimes and determine the concentration of DTT that induces the lethality of the Lys11Arg mutant.

Taken together, these data represent a step towards understanding the apparent dichotomy between the Ub conservation and the previously determined Ub fitness landscape. While much of

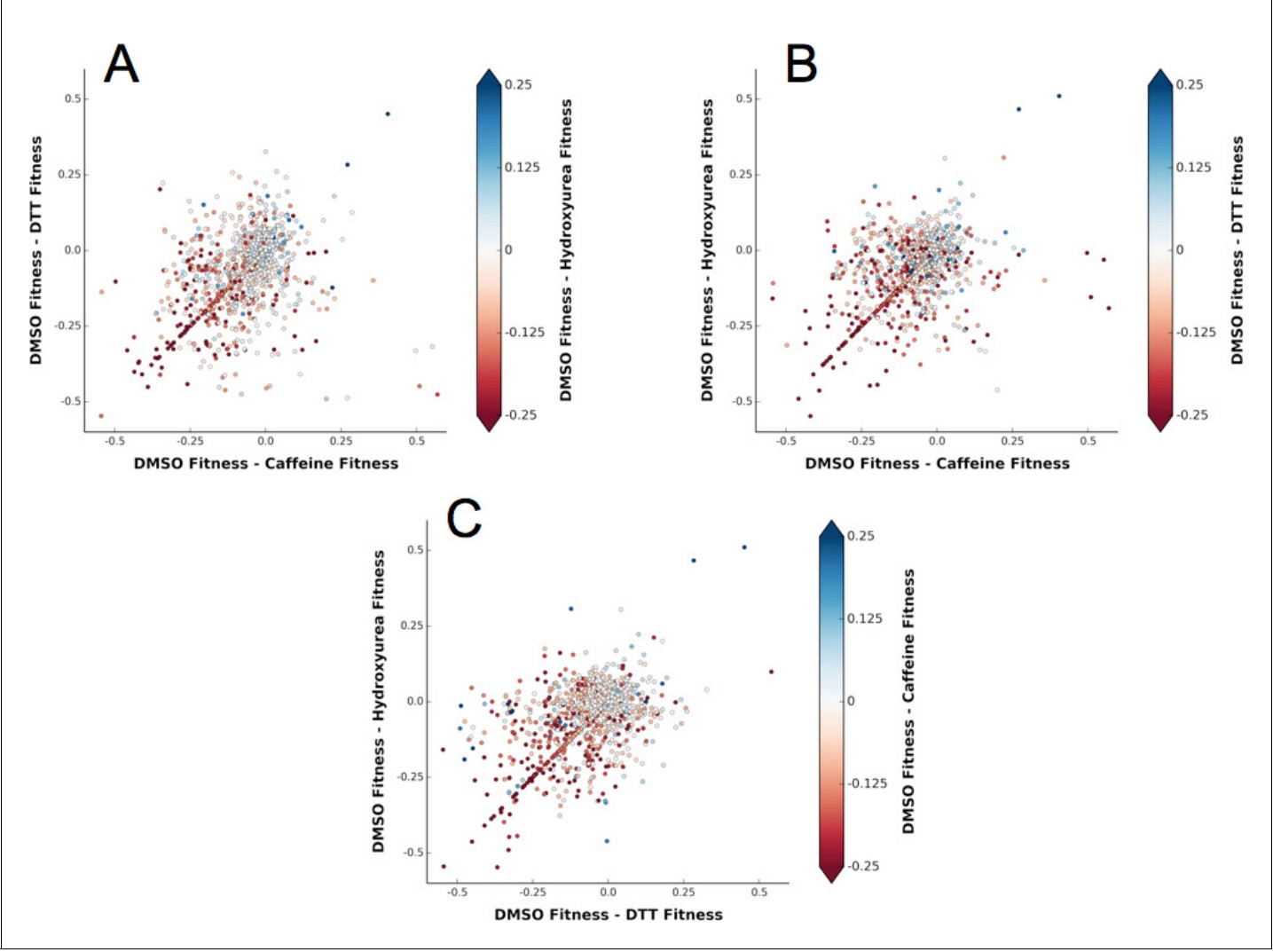

**Figure 9.** A shared response to different chemical perturbations. (**A**) DMSO fitness - Caffeine fitness vs. DMSO fitness - DTT fitness. The markers are colored based on DMSO fitness - Hydroxyurea fitness. (**B**) DMSO fitness - Caffeine fitness vs. DMSO fitness - Hydroxyurea fitness. The markers are colored based on DMSO fitness - DTT fitness. (**C**) DMSO fitness - DTT fitness vs. DMSO fitness - Hydroxyurea fitness. The markers are colored based on DMSO fitness - Caffeine fitness.

The following source data is available for figure 9:

**Source data 1.** Shared response mutants representing mutations that are equally perturbed by all three sensitizing perturbations.
**Source data 2.** Perturbation specific mutations represent alleles that are differentially affected by Caffeine, DTT and Hydroxyurea.
**Source data 3.** Specific information regarding highlighted mutants.

the protein is tolerant to mutation when cells are grown with traditional laboratory conditions, new stress conditions reveal hidden mutational sensitivity. We show that thirteen new positions are extremely sensitized in at least one stress condition with an additional thirteen new positions intermediately sensitized. While the incorporation of these new stresses provides a rationale for an additional 1/3 of the protein, we cannot currently explain the conservation of some positions in the 'tolerant' face of the protein. Expanding the set of chemical perturbations assayed may begin to address this dichotomy further. It is also possible that mutations at tolerant positions create fitness

defects that are too subtle to be determined by our current methods. These subtle defects can lead to the sequence conservation observed in Ub when a large population undergoes selection over a longer evolutionary time (*Boucher et al., 2014*). Future experiments may be able to identify these effects by: i) increasing the dose of the perturbation; ii) reducing the expression of the mutant Ub; or iii) performing the selection over more generations (*Rockah-Shmuel et al., 2015*).

The observed fitness defects can be due either to the functional properties of the mutant or the concentration of free Ub in the cell. The mutants in the library are all expressed on the same plasmid and the same promoter, which gave us confidence to interpret the effect of the mutants on Ub thermostablity and biological function. However, it is important to note that the fitness defects may be due to decreased Ub mutant expression or increased Ub mutant degradation.

Historically, many temperature sensitive mutants are not deficient in protein activity, but rather have increased protein turnover by the proteasome due to destabilizing mutations (*Gardner et al., 2005*). Ub turnover is unique in that free Ub is not degraded (*Shabek et al., 2007*) and conjugated Ub is degraded by the proteasome when the DUBs Upb6 and Rpn11 do not remove Ub from the substrate, causing Ub to be pulled into the proteasome along with the substrate protein (*Hanna et al., 2007*). Additionally, the free Ub pool can be increased by the activity of the DUB Doa4 (*Kimura et al., 2009*), which cleaves Ub conjugates into free Ub. Therefore some of the mutants may increase Ub turnover either by interfering with DUB recognition or by destabilizing Ub and easing Ub unraveling by the proteasome (*Lee et al., 2001*; *Prakash et al., 2004*). Selection experiments that probe protein-protein interactions more directly, similar to those performed between the yeast E1 and Ub (*Roscoe and Bolon, 2014*), may be able to directly determine whether some mutants have abnormal proteasomal engagement and turnover.

The inability of SUB328 to regulate the free Ub pool by increasing expression may also limit the interpretation of our results. Physiologically Ub protein levels are maintained by strong expression of the UBI4 locus in response to stress conditions (*Finley et al., 1987*). In the strain used in these experiments (SUB328 (*Finley et al., 1994*), the native Ub loci have been deleted and complemented with both a plasmid containing WT Ub and a plasmid containing the mutant Ub. Therefore, SUB328 can no longer respond to stress by increasing the expression of UBI4 and is entirely dependent on DUB based mechanisms to maintain the free Ub pool, increasing the fitness defects of mutants that interfere with Doa4, Upb6, and Rpn11 activity. Alternatively, SUB328 expressing a deficient Ub mutant might increase Ub expression by increasing the copy number of the plasmid. While integrating these Ub alleles into the genome would remove copy number variation, it would also dramatically decrease number of Ub variants that could be assessed due to the relative inefficiency of integration when compared to transformation.

While these caveats are a concern, the SUB328 experimental system has been successfully used for many years to assess the effect of Ub mutations in yeast (*Roscoe et al., 2013*; *Sloper-Mould et al., 2001*; *Lee et al., 2014*). Furthermore, we have only interpreted mutants that cause large defects, such as the biologically sensible fitness defects of mutations at known important residues such as Leu8 or Lys48 and residues with fitness values that vary between conditions such as His68. The fitness effects at these variable positions are not simply due to expression or turnover defects, which should decrease fitness uniformly across the chemical stresses. Therefore, mutants that affect protein turnover or expression are likely members of the 'shared response' mutants described above. In the case where a perturbation increased Ub turnover we would expect that most positions would be uniformly sensitized. Instead we observe a stereotyped pattern of sensation for the three sensitizing perturbations, suggesting that Ub turnover is similar for all three perturbations and the 'perturbation specific mutations' are independent of protein turnover and/or expression defects.

These experiments also demonstrate the success of graduate-level project based courses (*Vale et al., 2012*) as key components of a first-year curriculum. Our students were able to generate high quality data and useable computational pipelines during the 8 weeks of class time. These successes are notable because few students began the class with a background in both areas. By creating a project lab environment that encouraged team based learning and teaching, we enabled students to quickly acquire relevant skills within the context of an active research project. The wide variety of stress responses that Ub mediates and the vast chemical space that can be safely and economically addressed in a classroom make yeast and Ub ideal systems for continuing these studies. It

is our hope that other graduate programs can similarly offer project based classes in their curriculums and we will make our reagents available for use to further that goal.

## Materials and methods

Additional material is available on our website (www.fraserlab.com/pubs_2014) and GitHub (https://github.com/fraser-lab/PUBS2014). Raw Sequencing reads are made available via SRA (SRR3194828)

### Yeast library

Yeast strain SUB328 (MATa *lys2-801 leu2-3,2–112 ura3-52 his3-Δ200 trp1-1 ubi1-Δ1::TRP1 ubi2-Δ2:: ura3 ubi3-Δub-2 ubi4-Δ2::LEU2* (pUB146) (pUB100)) (*Finley et al., 1994*) was used, which expresses ubiquitin from a galactose-inducible promoter in pUB146. pUB100 expresses the Ubi1 tail. A library of ubiquitin genes was saturated with point mutations (*Roscoe et al., 2013*). Barcodes were added by ligating N18 oligos flanked by EagI and AscI sites into each of the eight previously create Ub libraries. These libraries were bottlenecked by transformation into *E. coli* and then pooled to create the single N18BC-UbLib. This pooled library was transformed into *E. coli* to create the final N18BC-UbLib.

### Barcode association PCR/library/sequencing

To associate the N18BCs to a given Ub allele, we performed a paired end read on the Illumina MiSeq. Because Ub is a small gene, we were able to read the entire ORF with a 260 bp read and the associated N18BC with a 30 bp read. To prepare the library for sequencing, plasmid DNA was extracted from E. coli using the Omega Bio-Tek mini-prep kit. A ~700 bp product was amplified with primers containing the Illumina PE1 and PE2 primer sequences for 9 cycles to minimize PCR recombination. These products were separated on agarose gel, and excised products were purified by silica column. This library was prepared for sequencing on the Illumina MiSeq.

### Drug concentration

The concentration to reduce the growth rate of SUB328 (WT Ub) by 25% was determined by monitoring the growth of cells by optical density measurements at 600nm over 8 hr. MG132 and DMSO did not affect SUB328 (WT Ub) growth rate at any tested concentration. Hydroxyurea treatment induces a lag-phase followed by WT like growth.

### EMPIRIC-BC

Transformation

SUB328 strain was independently transformed three times with the barcoded Ub library. Two of these transformations (LibA, LibB) were transformed with the LiAc method described previously (*Gietz and Woods, 2002*). The third library (LibC) was transformed using the hybrid LiAc/electroporation protocol described previously (*Benatuil et al., 2010*). Libraries were grown in log phase for 48 hr @ 30*C in SRGal (synthetic, 1% raffinose, 1% galactose) + G418 and ampicillin and then flash frozen in LN2 at late log phase and stored at -80*C as 1 mL aliquots.

Library growth and sample collection

Frozen aliquots were thawed and grown in 50 mL SRGal +G418 in log-phase for 48 hr to remove dominant negative alleles. The library was transferred into SD (synthetic, 2% glucose) + G418 as described (*Roscoe et al., 2013*). The library was maintained in log-phase for 12 hr in SD + G418, at which time an initial sample was collected as described (*Roscoe et al., 2013*). The libraries were then maintained in log-phase growth by diluting cells into fresh SD +G418 every 12 hr, in the presence of the perturbation. The perturbation was refreshed with each dilution. Samples were taken every 2–3 SUB328 (WT Ub) generations.

PCR and miniprep

Plasmid DNA was extracted from yeast and prepared for deep sequencing. Yeast pellets were thawed and lysed and plasmid DNA recovered as previously described (*Roscoe et al., 2013*). A 268 bp product was amplified from the plasmids by PCR, using only 9 cycles of amplification. This

product contained the N18BC. PCR products were separated on agarose gel, and excised products were purified by silica column. A second round of PCR was performed to add unique indices (Illumina TruSeq) to barcode each sample.

## Sequencing and data analysis

Each PCR product was quantified by qBit and diluted to 4 nM. The samples were then pooled and the pooled libraries prepared for sequencing on the Illumina HiSeq. The N18BCs were sequenced with a single end HiSeq run with a custom primer (TGCAGCGGCCCTGAGTCCTGCC) that read directly into the N18BC. Samples were indexed using the HiSeq indexing read and the Illumina TruSeq indices.

## Pipeline
### Module 0: Sub assembly
Script1:

01_sele_BC.py paired_end_read1.fasq > good_BC_reads.fastq

This script takes a raw fastq file (Illumina output) and checks each sequence to see if it matches the expected Ub-Library vector sequence after the N18 bar-code Input file should be the Read1 output file of a paired end Illumina read. The output is matched sequences in fastq format printed to the terminal. The script will also write a log file named 'Script01_logfile.txt' by default

02_pair_reads.py good_BC_reads.fastq paired_end_read2.fastq -o pair_dict.pkl

This script takes the output fastq from 01_sele_BC.py and creates a dictionary keyed on the sequence sample ID. It then takes the full raw read2 fastq and associates the sample N18 bar-code with the Ub sequence from read2. The output is a dictionary keyed on the sample ID with values as a 2 item list. the first entry is the N18 bar-code (pair_dict[identity_key][0]). The second is a list of the Ub sequence from read2 (pair_dict[identity_key][1]).

pair_dict.pkl

{'SampleID':['Barcode', 'Ub_sequence'], . . .}

03_sequences_assigned_to_barcode.py pair_dict.pkl -o barcode_to_Ub.pkl

This script takes the output from 02_pair_reads.py and associates a given N18 barcode with all the related ubiquitin sequences. It then returns a dictionary that is keyed on the barcode with values of a list of all associated ubiquitin reads.

barcode_to_Ub.pkl

{'Barcode': ['Ub_sequence1', 'Ub_sequence2', . . .]... }

04_generate_consensus.y barcode_to_Ub.pkl -o Allele_Dictionary.pkl

This script takes the output from 03_sequences_assigned_to_barcode.py and generates a consensus sequence from the list of Ub sequences associated with a given barcode. The mutant in the consensus sequence is identified and associated with the barcode. A barcode must be observed at least 3 times and the consensus sequence must contain only one mutation to be included. The output is a dictionary keyed on the barcode with a tuple value of (int(amino_acid_position), str(mutant_codon))

Allele_Dictionary.pkl

{"barcode": (aa_position_in_Ub, Mutant_Codon)}
"AGCTCTA": (74, AUU)
"AGCCCTA": (5, GCU)}

### Module 1: Extract BC counts from fastq with Hamming error correction

Requires seqmatch.py to be present in the working directory. The below scripts use function imported from this file

Script1:

fastq_index_parser_v4.py data.fastq –indices barcodes.txt -o indexed_data.pkl –index_cutoff 2 –const_cutoff 2

This will take the fastq files from a sequence run as input and create dictionaries that are keyed on the sample index and have values of barcode:quality score. The index and const cutoffs are Hamming distance cutoffs for the sample index (2 is acceptable because all TS BCs used are greater than 2 Hamming distance apart) and constant region of the vector (again 2 is acceptable because we are matching to a known constant region of length 8)

data.fastq - fastq formatted file directly from the sequencer

barcodes.txt - a table delimited file with 2 columns, the first being the name of the index and the second being the DNA sequence of the index

TS1 CGTGAT

TS2   ACATCG

TS3 GCCTAA

indexed_data.pkl

{TS1:{barcode:quality-score,... }, TS2:{barcode:quality-score,... },... ,}

{TS1:{'AGCTCTA':'*55CCF>',...}...}

Script2:

pkl_barcode_parser.py indexed_data.pkl –out_pickle indexed_data_counts.pkl –allele_pickle Allele_-Dictionary.pkl –fuzzy_cutoff 2

This scripts takes in the pkl file produced by the previous script for each sample and checks the fastq quality scores and matches the sequenced barcodes to those identified by the assembly of the library and counts the number of times a barcode is observed. This script also uses the Hamming distance between an observed barcode and members of the Allele_Dictionary to assign counts to previously observed barcodes even if a sequencing error occurred in a given read. The 'fuzzy_cutoff' parameter sets the max Hamming distance considered.

indexed_data_counts.pkl

{TS1:{barcode:number-of-reads,... }, TS2:{barcode:number-of-reads,... }, . . .}

{TS1:{'AGCTCTA': 147,...}...}

Script 3:

pickleread.py indexed_data_counts_1.pkl... indexed_data_counts_N.pkl –out_dir output_files/ –pkl_basename TS_ –allele_pickle Allele_Dictionary.pkl

This script takes the output from multiple runs of 'pkl_barcode_parser.py' and combines the counts. This will result in one dictionary for each sample index with barcodes sequence as key and the number of reads as values.

The output files (pkl) will be as follows (30 pkl files):

TS1:{barcode:number-of-reads,... }

TS2:{barcode:number-of-reads,... }

. . .

TSN:{barcode:number-of-reads,... }

## Module 2: Initial scoring - Barcodes, initial counts cutoff = 3

Script1:

pickle_condensing.py TS_1.pkl TS_2.pkl TS_3.pkl perturbation replicate -o Barcode_Counts.pkl

This script simply takes the counts from the dictionaries created by Module 0 Script 3 and combines them into a single dictionary that contains the counts for a given barcode for all 3 samples that describe an experiment. The perturbation and replicate inputs are used in naming the output dictionary.

Script2:

Score_BCs.py Barcode_Counts.pkl Allele_Dictionary.pkl time1 time2 -o Barcode_scores.pkl

Barcode_Counts.pkl

{"barcode": [count_sample1, count_sample2...]}

"AGCTCTA": [15, 3, 1]

"AGCCCTA": [222, 23, 21]}

This script takes a. pkl of a dictionary (Barcode_Counts.pkl) keyed on the sample barcodes with values of a list of counts at each time point. The scoring function uses these counts and scores them based on the time values (in WT generations) - the relative fitness is compared to wild type barcodes, which are distinguished in Allele_Dictionary.pkl. Output is a dictionary keyed on sample barcode with values of the fitness score. Fitness scores are determined by calculating the slope of the regression line of the three counts for each barcode. The score reported is $\log_2$(Mutant Slope/ WT Slope). Any barcode that is observed three or less times in the initial sample is not used in the fitness score calculation

Barcode_scores.pkl

{"barcode": float(Fitness_score)}

"AGCTCTA": -0.56
"AGCCCTA": -0.1}

## Module 3: Outlier detection and removal

toss_outliers.py Barcode_scores.pkl codon 4 -o clean_BCs.pkl

clean_BCs.pkl

{"dirty_barcodes": [(aa_position_in_Ub, Mutant_Codon, barcode) . . .]
"clean_barcodes": [(aa_position_in_Ub, Mutant_Codon, barcode) . . .]}
"dirty_barcodes": [(74, AUU, "AGCTCTA"), (21, CCC, "ACTTCTA") . . .]
"clean_barcodes": [(5, GCU, "AGCCCTA"), (21, UUU, "GCATTTC") . . .]}

This script compares the scores of barcodes mapping to the same allele. The median absolute deviation (MAD) is calculated for the barcodes that map to the same allele. Outlier barcodes are determined as scores that are greater than or equal to 1.5 times the interquartile range of the distribution and removed. The codon flag tells the script to compare all scores mapping to the same codon. The 4 flag sets the minimum number of barcodes before the MAD will be performed. The output is a pkl of barcodes to be kept in the dataset.

remove_bad_BCs.py Barcode_scores.pkl clean_BCs.pkl Allele_Dictionary.pkl -o Barcode_scores_outliers_removed.pkl

This script checks the Barcode_scores dictionary against the clean_BCs returned by toss_outliers. py and removes those BCs that are not in the clean_BCs.pkl. The script returns a dictionary keyed on sample barcode with fitness scores as values but with outlier BCs removed.

Barcode_scores_outliers_removed.pkl

{"barcode": float(Fitness_score)}
"AGCCCTA": -0.1
"GCATTTC": -0.67}

## Module 4: Create matrix

heatmap_BCs.py Barcode_scores_outliers_removed.pkl Allele_Dictionary.pkl -o Barcode_scores_outliers_removed_matrix.pkl

This script takes the Barcode scores and averages them to the amino acid level. It then outputs these scores as a heatmap and as a numpy matrix pkl.

Barcode_scores_outliers_removed_matrix.pkl

masked_array(data = [21X76 matrix containing fitness scores for each aa substitution])

[[– -0.631791406846642 -0.5397724613430753... , -0.3530569856099873
-0.4209070611721436 –]
[– -0.04155544432657808 –... , – -0.4672829444990562
-0.015863306980341812]
[– -0.06881685222913404 -0.3000996826283508... , -0.09038257104760622
-0.5060198247122988 –]
... ,
[– -0.2136845391374962 -0.5846954623554699... , -0.5201027046981986 – –]
[– -0.037103840372513595 -0.6445621511224743... , -0.6398418859301449
-0.5009308293341608 –]
[– -0.03813077561871194 -0.5946959237696324... , -0.5684471534238328
-0.3959407495759722 –]]

## Rosetta predictions of ubiquitin stability changes upon point mutations

We used Rosetta version number 55,534 for all simulations. The Rosetta software can be downloaded at www.rosettacommons.org.

Using the crystal structure of human ubiquitin (1UBQ) as input, we first introduced three mutations to match the *S. cerevisiae* sequence using Rosetta fixed backbone design:

**Command line:**

fixbb.linuxgccrelease -s 1UBQ.pdb -resfile UBQ_to_yeast.res -ex1 -ex2 -extrachi_cutoff 0 -nstruct 1 -overwrite -linmem_ig 10 -minimize_sidechains

**UBQ_to_yeast.res file contents**:

```
NATRO
start
19 A PIKAA S
24 A PIKAA D
28 A PIKAA S
```

We then followed a protocol described by Kellogg & coworkers (*Kellogg et al., 2011*) for estimating stability changes in monomeric proteins in response to point mutations. For documentation of the protocol and file formats (mut_file, cst_file), see https://www.rosettacommons.org/docs/latest/application_documentation/analysis/ddg-monomer

The first step minimizes the input structure (the model of yeast ubiquitin generated above, 1UBQ_0001.pdb):

**Command line** (weights file sp2_paper_talaris2013_scaled.wts in supplement):

```
minimize_with_cst.static.linuxgccrelease -s 1UBQ_0001.pdb -in:file:fullatom -ignore_unrecognized_res -fa_max_dis 9.0 -ddg::harmonic_ca_tether 0.5 -ddg::constraint_weight 1.0 -ddg::out_pdb_prefix min_cst_0.5 -ddg::sc_min_only false -score::bonded_params 300 150 40 40 40 -scale_d 1 -scale_theta 1 -scale_rb 1 -score:weights sp2_paper_talaris2013_scaled.wts
```

The second step performs the stability calculations:

**Command line:**

```
ddg_monomer.static.linuxgccrelease -in:file:s 1UBQ_minimized.pdb -ddg::mut_file (mutfile) -constraints::cst_file (cst_file) -ignore_unrecognized_res -in:file:fullatom -fa_max_dis 9.0 -ddg::dump_pdbs true -ddg::suppress_checkpointing true -ddg:weight_file soft_rep_design -ddg::iterations 50 -ddg::local_opt_only false -ddg::min_cst true -ddg::mean false -ddg::min true -ddg::sc_min_only false -ddg::ramp_repulsive true -score::bonded_params 300 150 40 40 40 -scale_d 1 -scale_theta 1 -scale_rb 1 -ddg:minimization_scorefunction sp2_paper_talaris2013_scaled.wts
```

## Acknowledgements

We acknowledge: administrative support from Rebecca Brown, Julia Molla, and Nicole Flowers; technical support from Jennifer Mann and Manny De Vera; gifts from David Botstein, and Illumina; and helpful discussions with Hana El-Samad, Nevan Krogan, Danielle Swaney, and Ron Vale. The Project Lab component of this work is specifically supported by an NIBIB T32 Training Grant, 'Integrative Program in Complex Biological Systems' (T32-EB009383). UCSF iPQB and CCB Graduate programs are supported by US National Institutes of Health (NIH) grants EB009383, GM067547, GM064337, and GM008284, HHMI/NIBIB (56005676), UCSF School of Medicine, UCSF School of Pharmacy, UCSF Graduate Division, UCSF Chancellors Office, and Discovery Funds. LSM, SDA, ST, and WC are supported by NSF Graduate Research Fellowships. EMG is supported by a Kellogg Chancellor Fellowship. ECS is supported by the Initiative for Maximizing Student Development program. DNB is supported by NIH GM112844. JSF is a Searle Scholar, Pew Scholar, and Packard Fellow, and is supported by NIH OD009180. TK is supported by NIH GM117189 and NSF DBI-1564692.

## Additional information

### Funding

| Funder | Grant reference number | Author |
| --- | --- | --- |
| National Science Foundation | Graduate Research Fellowships | Samuel Thompson<br>Seth D Axen<br>Weilin Chen<br>Leanna S Morinishi |
| University of California, San Francisco | Ralph H Kellogg Endowed Chancellor's Fellowship | Evan M Green |
| National Institute of General Medical Sciences | Initiative for Maximizing Student Development Program | Eugenia C Salcedo |
| National Institutes of Health | GM117189 | Tanja Kortemme |

| National Science Foundation | DBI-1564692 | Tanja Kortemme |
| National Institutes of Health | GM112844 | Daniel N Bolon |
| NIH Office of the Director | OD009180 | James S Fraser |

The funders had no role in study design, data collection and interpretation, or the decision to submit the work for publication.

## Author contributions

DM, Conception and design, Acquisition of data, Analysis and interpretation of data, Drafting or revising the article, Contributed unpublished essential data or reagents; KB, ST, BAB, JL, Conception and design, Acquisition of data, Analysis and interpretation of data, Drafting or revising the article; ARB, CLC, GG, ZL, LD, SDA, EC, WC, AC, REG, EMG, KRH, WJ, LRK, BM, LSM, SMM, MM, RKM, SN, CIN, EP, EMP, TDR, ECS, SKS, MS, AWW, Acquisition of data, Analysis and interpretation of data, Drafting or revising the article; KST, JLD, Conception and design, Drafting or revising the article; SÓC, Acquisition of data, Contributed unpublished essential data or reagents; BPR, Conception and design, Drafting or revising the article, Contributed unpublished essential data or reagents; EDC, Conception and design, Acquisition of data, Drafting or revising the article; TK, Conception and design, Analysis and interpretation of data, Drafting or revising the article; DNB, JSF, Conception and design, Analysis and interpretation of data, Drafting or revising the article, Contributed unpublished essential data or reagents

## Author ORCIDs

Kyle Barlow, http://orcid.org/0000-0002-9787-0066
James S Fraser, http://orcid.org/0000-0002-5080-2859

## Additional files

### Major datasets

The following dataset was generated:

| Author(s) | Year | Dataset title | Dataset URL | Database, license, and accessibility information |
| --- | --- | --- | --- | --- |
| Mavor D | 2016 | Deep mutational scan of Ub - Student Lane 1 | http://www.ncbi.nlm.nih.gov/sra/?term=SRR3194828 | Publicly available at the NCBI Short Read Archive (accession no: SRR3194828). |

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
