## [Decision Letter]

[Editors’ note: a previous version of this study was rejected after peer review, but the authors submitted for reconsideration. The previous decision letter after peer review is shown below.]

Thank you for submitting your work entitled "Determination of Ubiquitin Fitness Landscapes Under Different Chemical Stresses in a Classroom Setting" for consideration by *eLife*. Your article has been reviewed by two expert peer reviewers, and the evaluation has been overseen by a Reviewing Editor (Jeffery W. Kelly) and Diethard Tautz as the Senior Editor. Our decision has been reached after consultation between the reviewers. Based on these discussions and the individual reviews below, we regret to inform you that your work will not be considered further for publication in *eLife*.

The authors propose that the major take-home of their paper is to measure the fitness of all Ubiquitin mutants after exposing yeast to distinct stresses, i.e. chemical perturbations, the idea being that mutants might be functionally deficient under stress conditions-which is not a new concept.

As Reviewer 1 points out, the authors do not consider the possibility that mutations could affect the level (or turnover rate) of ubiquitin. Rather, the tendency is to attribute the fitness effects to perturbed protein-protein interactions, ubiquitin chain formation, and so forth-which could be the case, but, as in any mutagenesis study, effects on protein level or degradation are possible and it seems that this must be probed for this paper to be considered.

Reviewer 2 feels that the authors have only partially succeeded in resolving the apparent conflict between the strict conservation of ubiquitin sequence in evolution and its mutational tolerance in the lab. The authors reveal that different stresses do sensitize new positions in ubiquitin, but they fail to deal directly with the fundamental limitation of experimental assays to measure small fitness effects, which seems like a very important issue regarding drawing strong conclusions from this work.

This work is certainly an inspiration and a call-to-arms for graduate programs, since it demonstrates the potential of project-based courses. We only encourage resubmission if the authors can deal with all of the reviewer comments, especially the above-mentioned critical concerns of reviewers 1 and 2 (see below for the remainder of the reviewer comments).

*Reviewer #1:*

Since the 80's ubiquitin has been assaulted by structural biochemists, physical biochemists, and geneticists. It is of interest because of its central role in cellular regulation, its extraordinary evolutionary conservation, the multiplicity of its interactors, the stability of it structure, and the speed of its folding. The present study is motivated by the conundrum that ubiquitin is so highly conserved while in experimental settings tolerating substitution at many residues.

This paper describes a plunge into the importance of individual amino acid residues to fitness across the sequence of ubiquitin, focusing on stress conditions, using a very sophisticated approach including a deep dataset. It's a good rationale since stress states and particularly stationary phase (though that was not examined) probably do contribute substantially to natural selective pressures. The most interesting data are those showing distinct mutational profiles depending on the type of stress applied. In an extreme case, the stressor MG132 improved rather than attenuated mutational robustness. In general this is a well-executed and interesting study. Thus the data are good and I fully support publication. However, it may be that the interpretation is a bit narrow, a problem that is correctable.

The Discussion does not consider the possibility that mutations could affect the level (or turnover rate) of ubiquitin. Rather, the tendency is to attribute the fitness effects to perturbed protein-protein interactions, ubiquitin chain formation, and so forth – which may well be the case for the bulk of mutants. But in any mutagenesis, possible effects on protein level or degradation rates are in play, or to put it differently, this may play a role in the fitness landscape of many proteins. Many ts mutants are ts for protein levels, not function, with the ts protein being eliminated by the proteasome (Gardiner et al. Cell 120, 803 [2005]). Is ubiquitin any different? Ubiquitin is an unusual protein in these respects. Under stress conditions, ubiquitin rapidly declines in levels, and the UBI4 gene, running off of a strong promoter, serves principally to preserve ubiquitin levels under a variety of stress conditions, and in stationary phase (Cell 1987; 48: 1035-1046 [2007]). Other genes also play key roles in sensing and adjusting ubiquitin levels (Mol Cell Biol. 23: 9251-9261 [2003]; Cell 129, 747 [2007]; Cell 137: 549-59 [2009]). In the host strain, the levels of ubiquitin, if wild-type, do appear to be adequate after shift to galactose (Mol Cell Biol 23: 9251-9261 [2003]).

Why does ubiquitin decline under stress? In yeast, ubiquitin is degraded with a half-life of ~2 hr, as measured in this strain. This proceeds mainly via the proteasome (Mol Cell Biol 23: 9251-9261 [2003]), when deubiquitinating enzymes Ubp6 and Rpn11 together fail to remove ubiquitin from proteolytic substrates, and ubiquitin is dragged into the core particle. Accordingly, only conjugated ubiquitin is degraded, not free (Biochem Biophys Res Comm 363:425-31 [2007]).

Ubiquitin is exceptionally stable physically (for example its melting temperature is about 85°C) though not at all metabolically. The authors allude to ubiquitin stability but only in the sense of physical stability. Their in silico approach points to a role of physical stability in accounting for some of the observed effects but it's possible that the physical stability effects are mediated through metabolic instability, which was not examined, or that metabolic stability effects are in play where physical stability is only modestly perturbed.

The exceptional physical stability may reflect the need to minimize ubiquitin degradation by the proteasome, an occupational hazard unique to this molecule. If the proteasome fails to unfold ubiquitin, ubiquitin will not be dragged into the core particle and degraded, but rather it will be removed from substrate by deubiquitinating enzymes. This need for ubiquitin to resist the proteasome's mechanical unfolding action could possibly underlie the exceptional physical stability of ubiquitin and some component of its sequence conservation, whether substantial or not. Put differently, it might be reflected to some extent in the fitness landscape of ubiquitin. However, as noted above, unfolding by the proteasome is not equivalent to thermal unfolding (Mol Cell 7:627-37 [2001]; Nat Struct Mol Biol 11:830-7 [2004]), so purely physical studies might miss relevant effects.

Since all endogenous ubiquitin genes are deleted in the host strain, some of the ability to regulate ubiquitin levels is also lost in the strain, and cells may consequently have trouble adapting to stress. Whether this is affecting results in these particular cases I don't know, but it is worth considering or mentioning. Ideally one would assess ubiquitin levels in the various mutants, though because they are mutants antibody recognition may be affected, so this approach could be flawed. Quantitative mass spectrometry would be the preferred method.

The importance of ubiquitin levels could be underestimated in work involving steady-state growth in the host strain, because any effect on ubiquitin levels can be compensated by adjustments in plasmid copy number. This is not a fast mechanism so would essentially not operate over the time course of the present studies. Thus, when examining ubiquitin under stress, in this short-term protocol, this putative effect could be more prominent than otherwise. Ultimately, integrated mutants will have advantages where ubiquitin levels are a concern, as potential plasmid copy number effects will be eliminated. It may be argued that the different relative mutant fitness seen under different stresses mitigates ubiquitin levels as a concern. Perhaps so, but degradation itself will be affected differently by these stresses; I don't think this argument can entirely neutralize the problem.

One of the more interesting findings in the paper is that the MG132-specific effects are overall opposite in sign to those of other stressors. High dose MG132 drives ubiquitin into conjugates and depletes free pools. For low doses such as used here, who knows. Conceivably the present results may be explained by an overall slowing of ubiquitin degradation, thus preserving ubiquitin levels over the time course of the experiment.

*Reviewer #2:*

The authors present the acquisition and analysis of ubiquitin fitness landscapes under a set of distinct chemical stresses. The main goal of the work is to resolve the apparent conflict between the strict conservation of ubiquitin sequence in evolution and its mutational tolerance in the lab. In this task the authors are only partially successful, revealing that different stresses do sensitize new positions in ubiquitin but failing to deal directly with the fundamental limitation of experimental assays to measure small fitness effects. I found the experiments and analysis to be well executed, although details are lacking in some areas. The work is an inspiration and a call-to-arms for graduate programs, since it demonstrates the potential of project-based courses.

General comments:

1) I don't like the bone white -> blue color scheme, especially for the scatter plots. Because the bone white is so close to white, points on this end of the spectrum are hard to see and generally contribute much less visual weight to the plot. Such a color scheme would make sense if the information being conveyed was, for example, a p-value (such that points with small values should carry less weight than points with large ones). However, they're equally important so the authors should change their color scheme.

2) The authors make the argument that there is a paradox inherent in the mutational tolerance of ubiquitin and its sequence conservation. They propose that distinct stressors place distinct constraints on ubiquitin and that integrating over all relevant stressors produces strong constraint. In the Discussion, they briefly consider a very important alternative hypothesis: that the assays used to measure ubiquitin mutational tolerance in the EMPIRIC and ala-scan papers are insensitive to small fitness effects. Large populations and evolutionary timescales can act on such small fitness effects. This caveat should be mentioned earlier on. I also think the manuscript could benefit from an expanded discussion of this point. To what extent can experiments like these address this question? How does this study relate to others in this area (e.g. PMID: 26274323)?

3) The idea that lab conditions mask constraints on protein sequence (and genomes generally) is not new. I was disappointed to find that the manuscript lacked a discussion of other work in this broad area and that the authors did not place their work in the context what is already known. My impression is that the work, while providing some interesting insights into specific ubiquitin mutations, didn't have much to add to the general problem of mutational context dependence.

4) The authors include insufficient detail, often using imprecise language and failing to provide quantitative assessments. It really limited my enthusiasm for the work.

5) The authors should make the raw reads available on SRA. The analyzed data should also be made available.

---

## [Author Response]

[Editors’ note: the author responses to the previous round of peer review follow.]

The authors propose that the major take-home of their paper is to measure the fitness of all Ubiquitin mutants after exposing yeast to distinct stresses, i.e. chemical perturbations, the idea being that mutants might be functionally deficient under stress conditions-which is not a new concept.

As Reviewer 1 points out, the authors do not consider the possibility that mutations could affect the level (or turnover rate) of ubiquitin. Rather, the tendency is to attribute the fitness effects to perturbed protein-protein interactions, ubiquitin chain formation, and so forth-which could be the case, but, as in any mutagenesis study, effects on protein level or degradation are possible and it seems that this must be probed for this paper to be considered.

Reviewer 2 feels that the authors have only partially succeeded in resolving the apparent conflict between the strict conservation of ubiquitin sequence in evolution and its mutational tolerance in the lab. The authors reveal that different stresses do sensitize new positions in ubiquitin, but they fail to deal directly with the fundamental limitation of experimental assays to measure small fitness effects, which seems like a very important issue regarding drawing strong conclusions from this work.

This work is certainly an inspiration and a call-to-arms for graduate programs, since it demonstrates the potential of project-based courses. We only encourage resubmission if the authors can deal with all of the reviewer comments, especially the above-mentioned critical concerns of reviewers 1 and 2 (see below for the remainder of the reviewer comments).

We thank the reviewers and editors for their consideration of our manuscript. Since the major changes requested were to include caveats to the reductionist framework of attributing all fitness effects to changes of the folded ubiquitin protein (reviewer 1) and to interpreting small fitness changes (reviewer 2), we think that we have dealt with all the comments, especially the critical concerns. We apologize for the delay in returning the revised manuscript, especially in light of the fact that all the changes are textual and did not require new experiments. In addition to the usual excuses, we were running this year’s version of the class and wanted to wait until after it was complete before resuming this manuscript.

Reviewer #1:

*Since the 80's ubiquitin has been assaulted by structural biochemists, physical biochemists, and geneticists. It is of interest because of its central role in cellular regulation, its extraordinary evolutionary conservation, the multiplicity of its interactors, the stability of it structure, and the speed of its folding. The present study is motivated by the conundrum that ubiquitin is so highly conserved while in experimental settings tolerating substitution at many residues. This paper describes a plunge into the importance of individual amino acid residues to fitness across the sequence of ubiquitin, focusing on stress conditions, using a very sophisticated approach including a deep dataset. It's a good rationale since stress states and particularly stationary phase (though that was not examined) probably do contribute substantially to natural selective pressures. The most interesting data are those showing distinct mutational profiles depending on the type of stress applied. In an extreme case, the stressor MG132 improved rather than attenuated mutational robustness. In general this is a well-executed and interesting study. Thus the data are good and I fully support publication. However, it may be that the interpretation is a bit narrow, a problem that is correctable.* We thank the reviewer for the kind words and have worked to broaden the interpretation. We have added the following text to the Discussion:

“The observed fitness defects can be due either to the functional properties of the mutant or the concentration of free Ub in the cell. The mutants in the library are all expressed on the same plasmid and the same promoter, which gave us confidence to interpret the effect of the mutants on Ub thermostability and biological function. However, it is important to note that the fitness defects may be due to decreased Ub mutant expression or increased Ub mutant degradation. […] While these caveats are a concern, the SUB328 experimental system has been successfully used for many years to assess the effect of Ub mutations in yeast (Roscoe et al., 2013; Sloper-Mould et al., 2001; Lee et al., 2014). […] Instead we observe a stereotyped pattern of sensation for the three sensitizing perturbations, suggesting that Ub turnover is similar for all three perturbations and the “perturbation specific mutations” are independent of protein turnover and/or expression defects.”

The Discussion does not consider the possibility that mutations could affect the level (or turnover rate) of ubiquitin. Rather, the tendency is to attribute the fitness effects to perturbed protein-protein interactions, ubiquitin chain formation, and so forth – which may well be the case for the bulk of mutants.

As noted above, we have added the following text to the Discussion:

“The observed fitness defects can be due either to the functional properties of the mutant or the concentration of free Ub in the cell. The mutants in the library are all expressed on the same plasmid and the same promoter, which gave us confidence to interpret the effect of the mutants on Ub thermostability and biological function. However, it is important to note that the fitness defects may be due to decreased Ub mutant expression or increased Ub mutant degradation.”

But in any mutagenesis, possible effects on protein level or degradation rates are in play, or to put it differently, this may play a role in the fitness landscape of many proteins. Many ts mutants are ts for protein levels, not function, with the ts protein being eliminated by the proteasome (Gardiner et al. Cell 120, 803 [2005]).

As noted above, we have added the following text to the Discussion:

“Historically, many temperature sensitive mutants are not deficient in protein activity, but rather have increased protein turnover by the proteasome due to destabilizing mutations (Gardner et al., 2005).”

Is ubiquitin any different? Ubiquitin is an unusual protein in these respects. Under stress conditions, ubiquitin rapidly declines in levels, and the UBI4 gene, running off of a strong promoter, serves principally to preserve ubiquitin levels under a variety of stress conditions, and in stationary phase (Cell 1987; 48: 1035-1046 [2007]).

As noted above, we have added the following text to the Discussion:

“The inability of SUB328 to regulate the free Ub pool by increasing expression may also limit the interpretation of our results. Physiologically Ub protein levels are maintained by strong expression of the UBI4 locus in response to stress conditions (Finley et al., 1987).”

Other genes also play key roles in sensing and adjusting ubiquitin levels (Mol Cell Biol. 23: 9251-9261 [2003]; Cell 129, 747 [2007]; Cell 137: 549-59 [2009]).

As noted above, we have added the following text to the Discussion:

“…conjugated Ub is degraded by the proteasome when the DUBs Upb6 and Rpn11 do not remove Ub from the substrate, causing Ub to be pulled into the proteasome along with the substrate protein (Hanna et al., 2007). Additionally, the free Ub pool can be increased by the activity of the DUB Doa4 (Kimura et al., 2009), which cleaves Ub conjugates into free Ub.”

*In the host strain, the levels of ubiquitin, if wild-type, do appear to be adequate after shift to galactose (Mol Cell Biol 23: 9251-9261 [2003]).* As noted above, we have added the following text to the Discussion:

“In the strain used in these experiments (SUB328 (Finley et al., 1994), the native Ub loci have been deleted and complemented with both a plasmid containing WT Ub and a plasmid containing the mutant Ub…While these caveats are a concern, the SUB328 experimental system has been successfully used for many years to assess the effect of Ub mutations in yeast (Roscoe et al., 2013; Sloper-Mould et al., 2001; Lee et al., 2014).”

*Why does ubiquitin decline under stress? In yeast, ubiquitin is degraded with a half-life of ~2 hr, as measured in this strain. This proceeds mainly via the proteasome (Mol Cell Biol 23: 9251-9261 [2003]), when deubiquitinating enzymes Ubp6 and Rpn11 together fail to remove ubiquitin from proteolytic substrates, and ubiquitin is dragged into the core particle. Accordingly, only conjugated ubiquitin is degraded, not free (Biochem Biophys Res Comm 363:425-31 [2007]).* As noted above, we have added the following text to the Discussion:

“Ub turnover is unique in that free Ub is not degraded (Shabek et al., 2007) and conjugated Ub is degraded by the proteasome when the DUBs Upb6 and Rpn11 do not remove Ub from the substrate, causing Ub to be pulled into the proteasome along with the substrate protein (Hanna et al., 2007).”

*Ubiquitin is exceptionally stable physically (for example its melting temperature is about 85°C) though not at all metabolically. The authors allude to ubiquitin stability but only in the sense of physical stability. Their in silico approach points to a role of physical stability in accounting for some of the observed effects but it's possible that the physical stability effects are mediated through metabolic instability, which was not examined, or that metabolic stability effects are in play where physical stability is only modestly perturbed. The exceptional physical stability may reflect the need to minimize ubiquitin degradation by the proteasome, an occupational hazard unique to this molecule. If the proteasome fails to unfold ubiquitin, ubiquitin will not be dragged into the core particle and degraded, but rather it will be removed from substrate by deubiquitinating enzymes. This need for ubiquitin to resist the proteasome's mechanical unfolding action could possibly underlie the exceptional physical stability of ubiquitin and some component of its sequence conservation, whether substantial or not. Put differently, it might be reflected to some extent in the fitness landscape of ubiquitin. However, as noted above, unfolding by the proteasome is not equivalent to thermal unfolding (Mol Cell 7:627-37 [2001]; Nat Struct Mol Biol 11:830-7 [2004]), so purely physical studies might miss relevant effects.* As noted above, we have added the following text to the Discussion:

“Therefore some of the mutants may increase Ub turnover either by interfering with DUB recognition or by destabilizing Ub and easing Ub unraveling by the proteasome (Lee et al., 2001; Prakash et al., 2004)”.

*Since all endogenous ubiquitin genes are deleted in the host strain, some of the ability to regulate ubiquitin levels is also lost in the strain, and cells may consequently have trouble adapting to stress. Whether this is affecting results in these particular cases I don't know, but it is worth considering or mentioning. Ideally one would assess ubiquitin levels in the various mutants, though because they are mutants antibody recognition may be affected, so this approach could be flawed. Quantitative mass spectrometry would be the preferred method.* The inability to regulate Ub levels is a caveat that is now explained in the Discussion in the section starting:

“The inability of SUB328 to regulate the free Ub pool by increasing expression may also limit the interpretation of our results”.

Although the control suggested by the reviewer is intriguing, we think that it lies beyond the scope of this manuscript. We explored this suggestion seriously in consultation with mass spectrometry experts nearby. In order to quantify ubiquitin mutant expression with targeted MS we would follow the intensities of two well behaved ubiquitin peptides, TITLEVESSDTIDNVK (residues 12 to 27) and ESTLHLVLR (residues 64 to 72). We would be unable to quantify the expression of any mutant in those regions as the mutant would affect the flight of the peptide, making quantitative MS impossible. To normalize our peptide intensities to the number of cell in each sample, we would also have to follow the intensities of various peptides from housekeeping proteins that should have stable expression levels. To accomplish this task, we would first individually clone each point mutant into SUB328 and then grow each strain individually in glucose to repress the WT copy until the WT ubiquitin was removed by cell division. We would then lyse the cells and digest the lysate with trypsin and analyze the resulting peptides by MS. The time to clone each mutant and analyze the resulting samples by MS is significant and we think it would represent a separate contribution from the present work

The importance of ubiquitin levels could be underestimated in work involving steady-state growth in the host strain, because any effect on ubiquitin levels can be compensated by adjustments in plasmid copy number. This is not a fast mechanism so would essentially not operate over the time course of the present studies. Thus, when examining ubiquitin under stress, in this short-term protocol, this putative effect could be more prominent than otherwise. Ultimately, integrated mutants will have advantages where ubiquitin levels are a concern, as potential plasmid copy number effects will be eliminated.

As noted above, we have added the following text to the Discussion:

“Alternatively, SUB328 expressing a deficient Ub mutant might increase Ub expression by increasing the copy number of the plasmid. While integrating these Ub alleles into the genome would remove copy number variation, it would also dramatically decrease number of Ub variants that could be assessed due to the relative inefficiency of integration when compared to transformation.”

*It may be argued that the different relative mutant fitness seen under different stresses mitigates ubiquitin levels as a concern. Perhaps so, but degradation itself will be affected differently by these stresses; I don't think this argument can entirely neutralize the problem.* As noted above, we have added the following text to the Discussion:

“Furthermore, we have only interpreted mutants that cause large defects, such as the biologically sensible fitness defects of mutations at known important residues such as Leu8 or Lys48 and residues with fitness values that vary between conditions such as His68. […] Instead we observe a stereotyped pattern of sensation for the three sensitizing perturbations, suggesting that Ub turnover is similar for all three perturbations and the “perturbation specific mutations” are independent of protein turnover and/or expression defects.

*One of the more interesting findings in the paper is that the MG132-specific effects are overall opposite in sign to those of other stressors. High dose MG132 drives ubiquitin into conjugates and depletes free pools. For low doses such as used here, who knows. Conceivably the present results may be explained by an overall slowing of ubiquitin degradation, thus preserving ubiquitin levels over the time course of the experiment.* We have added the following text to the Results section:

“Also, reducing proteasome activity may increase the free pool of Ub in the cell by reducing the number of Ub proteins degraded by the proteasome. This increased pool of Ub could buffer the effects of deleterious Ub mutants participating in non- proteasomal functions. The consequence of an increased pool of free Ub might therefore lead to the general alleviating interactions observed between Ub mutants and MG132 treatment.”

Reviewer #2:

*The authors present the acquisition and analysis of ubiquitin fitness landscapes under a set of distinct chemical stresses. The main goal of the work is to resolve the apparent conflict between the strict conservation of ubiquitin sequence in evolution and its mutational tolerance in the lab. In this task the authors are only partially successful, revealing that different stresses do sensitize new positions in ubiquitin but failing to deal directly with the fundamental limitation of experimental assays to measure small fitness effects. I found the experiments and analysis to be well executed, although details are lacking in some areas. The work is an inspiration and a call-to-arms for graduate programs, since it demonstrates the potential of project-based courses. General comments: 1) I don't like the bone white -> blue color scheme, especially for the scatter plots. Because the bone white is so close to white, points on this end of the spectrum are hard to see and generally contribute much less visual weight to the plot. Such a color scheme would make sense if the information being conveyed was, for example, a p-value (such that points with small values should carry less weight than points with large ones). However, they're equally important so the authors should change their color scheme.*

We have changed the color scheme to go from red to blue with WT fitness set to white for all fitness scores.

*2) The authors make the argument that there is a paradox inherent in the mutational tolerance of ubiquitin and its sequence conservation. They propose that distinct stressors place distinct constraints on ubiquitin and that integrating over all relevant stressors produces strong constraint. In the Discussion, they briefly consider a very important alternative hypothesis: that the assays used to measure ubiquitin mutational tolerance in the EMPIRIC and ala-scan papers are insensitive to small fitness effects. Large populations and evolutionary timescales can act on such small fitness effects. This caveat should be mentioned earlier on. I also think the manuscript could benefit from an expanded discussion of this point. To what extent can experiments like these address this question? How does this study relate to others in this area (e.g. PMID: 26274323)?*We have added the following text to the Introduction:

“Small fitness defects that are undetectable by EMPIRIC may still be significant over evolutionary timescales and are likely to similarly be enriched for certain chemicals (Boucher et al., 2014). We suggest that expanding the set of environmental stresses, and improved measurement of smaller fitness defects, might be able to explain the high sequence conservation of ubiquitin, as different positions in the protein are important for interactions mediating the specific responses to a wide variety of perturbations.”

We have added the following text to the Discussion:

“It is also possible that mutations at tolerant positions create fitness defects that are too subtle to be determined by our current methods. […] Future experiments may be able to identify these effects by: i) increasing the dose of the perturbation; ii) reducing the expression of the mutant Ub; or iii) performing the selection over more generations (Rockah-Shmuel et al., 2015).”

3) The idea that lab conditions mask constraints on protein sequence (and genomes generally) is not new. I was disappointed to find that the manuscript lacked a discussion of other work in this broad area and that the authors did not place their work in the context what is already known. My impression is that the work, while providing some interesting insights into specific ubiquitin mutations, didn't have much to add to the general problem of mutational context dependence.

We have added the following text and references discussing the general problem of mutational context dependence:

“Previous EMPIRIC experiments on HSP90 suggested that reducing protein expression could reveal fitness defects that are otherwise buffered (Jiang et al., 2013). […] Collectively, these results suggest that the sequence of Ub is subject to many constraints arising from interacting with diverse proteins while mediating the stress responses to distinct chemical perturbations.”

4) The authors include insufficient detail, often using imprecise language and failing to provide quantitative assessments. It really limited my enthusiasm for the work.

We have addressed all specific comments about the language used and have quantified the relationship between our data and the previously published non-perturbed dataset. Additionally we have used the standard deviation of the fitness scores of each barcode that contributes to a mutant fitness score to estimate the error in each of our fitnesses.

5) The authors should make the raw reads available on SRA. The analyzed data should also be made available.

We have placed the raw reads on SRA: BioProject – PRJNA311570 BioSample – SAMN04485883

SRA – SRR3194828

We have also included CSVs of the floored fitness scores for each condition in our Github repository and could be included in the manuscript as supplementary tables.